



# Variability of air mass transport from the boundary layer to the Asian monsoon anticyclone

Matthias Nützel[1], Sabine Brinkop[1], Martin Dameris[1], Hella Garny[1,2], Patrick Jöckel[1], Laura L. Pan[3], and Mijeong Park[3]

[1]Deutsches Zentrum für Luft- und Raumfahrt, Institut für Physik der Atmosphäre, Oberpfaffenhofen, Germany
[2]Meteorologisches Institut München, Ludwig-Maximilians-Universität München, Munich, Germany
[3]National Center for Atmospheric Research, Boulder, Colorado, USA

**Correspondence:** Matthias Nützel (matthias.nuetzel@dlr.de)

**Abstract.** Air masses within the Asian monsoon anticyclone (AMA) show anomalous signatures in various trace gases. In this study, we analyze how air masses are transported from the planetary boundary layer (PBL) to the AMA via multiannual trajectory anlyses. While previous studies analyzed the PBL to AMA transport mainly for individual monsoon seasons or particular periods, we focus on the climatological perspective and on the interannual and intraseasonal variability.

To this end we employ backward trajectories, which were computed using reanalysis data. Based on these trajectories, we analyze air mass transport from the PBL to the AMA during northern summer (June–August) for 14 summer seasons. Further, we backtrack forward trajectories from a free-running chemistry-climate model (CCM) simulation, which includes parametrized Lagrangian convection. The analysis of this additional model data set helps us to carve out robust or sensitive features of PBL to AMA transport with respect to the employed model.

Results from both the trajectory model and the Lagrangian CCM emphasize the robustness of the three-dimensional transport pathways from the PBL to the AMA. Air masses are transported upwards on the eastern side of the AMA and are uplifted within the full AMA domain above. While this is in agreement with previous modelling studies, we refine the picture of the so-called "conduit" (Bergman et al., 2013). The contributions from the Tibetan Plateau (TP; 17% vs. 15%) and the West Pacific (around 12%) are similar in both model results. However, the contributions from the Indian subcontinent and South-East Asia are

considerably larger in the Lagrangian CCM data, which might point towards the importance of convective transport for PBL to AMA transport for these regions.

     The analysis of both model data sets highlights the interannual and intraseasonal variability with respect to PBL source regions of the AMA. Additionally, we analyze the relation of the interannual east–west displacement of the AMA - which we find to be related to the monsoon Hadley index - to the transport behaviour and find that there are differences for "east" and

"west years", the main transport characteristics, however, are comparable.

     Regarding the intraseasonal variability our trajectory model results show that transport from the PBL over the Tibetan Plateau (TP) to the AMA is weak in early June (less than 4% of the AMA air masses), whereas in August TP air masses contribute considerably (roughly 24%). The evolution of the contribution from the TP is supported by data from the Lagrangian CCM and is related to the northward shift of the subtropical jet and the AMA during this period. This result may help to reconcile previous



results and further highlights the need of taking the subseasonal (and interannual) variability of the AMA and associated transport into account.

# 1    Introduction

Strong precipitation during local summer is a typical criterion to define/identify monsoon regions (e.g. Wang et al., 2020). In the Asian summer monsoon (ASM) region heating related to the monsoon precipitation produces an anticyclone in the upper
troposphere and lower stratosphere (UTLS) over Asia (e.g. Hoskins and Rodwell, 1995; Park et al., 2007; Siu and Bowman, 2019, and references therein), which is often referred to as Asian (summer) monsoon anticyclone (AMA; e.g. Randel and Park, 2006; Park et al., 2007; Siu and Bowman, 2020).

Due to fast uplift of polluted air masses in the ASM region (von Hobe et al., 2021) and confinement within the AMA (Legras and Bucci, 2020), trace gases such as carbon monoxide (CO) show a maximum within the anticyclone (e.g. Santee
et al., 2017). Air masses that have reached the AMA or its edge can be further transported to the extratropical UTLS or the tropical stratosphere (e.g. Dethof et al., 1999; Randel et al., 2010; Vogel et al., 2014; Garny and Randel, 2016; Ploeger et al., 2017; Nützel et al., 2019; Vogel et al., 2019). In the stratosphere, these air masses might cause changes of the chemical and aerosol composition and hence affect the radiation budget (Randel et al., 2010). Thus, it is crucial to understand how trace gas anomalies within the AMA build up and how they are redistributed.
A first step towards answering these questions is to analyze the transport properties of air masses from the planetary boundary layer (PBL) to the AMA. This topic has been investigated in a couple of previous trajectory-based studies, e.g. by Bergman et al. (2013), Heath and Fuelberg (2014), Vogel et al. (2015), Fan et al. (2017), Vogel et al. (2019), Bucci et al. (2020) and Legras and Bucci (2020), sometimes with a focus on transport to the UTLS in the ASM region in general. All of these studies focus on individual important aspects regarding the transport to the AMA or UTLS in the ASM region.
As an example, Bergman et al. (2013) found a favourable region of upward transport on the eastern side of the AMA and coined the term of the so called conduit. Further, they calculated sensitivities with respect to the choice of the meteorological data used. Heath and Fuelberg (2014) focused on simulated high-resolution data to investigate the impact of rapid vertical transport to the AMA. Both of these studies highlighted the importance of the Tibetan Plateau with respect to transport from the PBL to the AMA. During the monsoon season 2017 comprehensive flight measurements have been conducted in the core of
the AMA within the StratoClim campaign (Bucci et al., 2020). Related to the flight campaign, two trajectory studies assessed the transport mechanisms and source regions of the air masses within the AMA in 2017: Bucci et al. (2020) analyzed the PBL source regions of air masses along the flight tracks to determine the source regions of the in-situ sampled air masses. Legras and Bucci (2020) studied the transport properties to and within the AMA and came to the conclusion that the conduit is driven by convection, whereas further ascent is driven by the large scale anticyclonic circulation. This finding is also in agreement
with the upward circling in the UTLS, which follows the first rapid ascent in the AMA region, as diagnosed by Vogel et al. (2019).



Despite these previous efforts, there is still a lack regarding the climatological picture and the description of the interannual and subseasonal variability of PBL to AMA transport. The typical short term or single season analysis presented in previous studies need to be tested for robustness, in particular if one considers the strong interannual and intraseasonal variability of the
AMA (e.g. Randel and Park, 2006; Garny and Randel, 2013; Siu and Bowman, 2020, and references therein) and of the whole monsoon system (e.g. Krishnamurti and Bhalme, 1976; Ding, 2007).

There are previous modelling studies, e.g. by Chen et al. (2012) and Fan et al. (2017), that looked into a multiannual analysis in the ASM region. However, these studies did not explicitly focus on transport from the PBL to the AMA but rather to a broad ASM region in the UT. As observations (apart from otherwise limited satellite data) are still rather scarce in the AMA region
(Brunamonti et al., 2018) and cannot directly provide information on the source region contributions, modelling studies are key to provide a climatological perspective of PBL to AMA transport without temporal or spatial gaps.

One example of the interannual variability of the AMA is the interannual variation of the east–west displacement of the center of the AMA (Wei et al., 2014). Wei et al. (2014) found a relation of enhanced Indian summer monsoon precipitation to the westward displacement of the AMA, which is supported by their simplified modelling studies (see also Wei et al., 2015, for
further analyses on the interannual variability of the AMA). Anomalous vertical wind fields in the UTLS over the ASM region corresponding to the longitudinal location of the AMA were shown by Nützel et al. (2016, their Fig. 14). This finding points toward a possible relation of the east–west displacement of the AMA with the transport characteristics in the ASM.

With respect to the intraseasonal variability, Vogel et al. (2015) found a strong variability in the source region contributions to the AMA at 380 K during the monsoon season 2012. This result highlights the need to assess the evolution of the source
regions of the AMA air masses during the course of the monsoon season in more detail.

With this additional viewpoint, we aim to bring together results of previous analyses and to add to the understanding of the composition of the AMA. The key questions we want to address are:

1. What is the climatological perspective of PBL to AMA transport in terms of pathways and PBL source regions? How reliable are previous results?

2. How do the pathways and source regions vary on intraseasonal and interannual time scales?

3. Are the PBL source regions and the transport pathways related to interannual east-west shifts of the AMA?

Our main focus lies on the analysis of backward-trajectories, which start in the core of the AMA, are driven by reanalysis data and are followed backward in time to their PBL origin. Further, the results from the trajectory analyses will be discussed with additional analyses from chemistry-climate model (CCM) simulations with a Lagrangian transport model. In particular, the
Lagrangian CCM results are from a free-running simulation and include the impact of parametrized Lagrangian convection. Results from this model will serve as a sensitivity in comparison to the reanalysis-based backward trajectory results as (i) (parametrized Lagrangian) convection, (ii) a different large scale dynamical background and (iii) forward trajectories (analyzed backward in time) are considered. This will help us to carve out key features that are similar or sensitive to the different modelling approaches. Further, the multiannual Lagrangian CCM data allow for additional analyses to complement the findings
in the trajectory model data.





The remainder of this study is structured as follows: Section 2 describes the data and methods used, including short descriptions of the employed models. The section thereafter (Sect. 3) contains the results of the trajectory simulations. These are complemented by the results of the Lagrangian CCM simulations in Sect. 4 and discussed in Sect. 5. Finally, we summarize our findings and state our concluding remarks (Sect. 6).

## 2   Data and method

In this study, we mainly focus on the analysis of data from a trajectory model to investigate the transport from the PBL to the AMA. The trajectory model propagates a set of trajectories, which are initialized by the user, using meteorological data e.g. from reanalysis data sets. Details on the trajectory model setup will be described in the next subsection and the reanalysis data is described at the end of this section.

Further, data from a CCM including a Lagrangian transport model (Brinkop and Jöckel, 2019), which features a Lagrangian convection parametrization, will be employed to complement the trajectory model results. This modelling approach differs from the trajectory model, e.g. as air parcels of the Lagrangian transport model are initialized only once at the start of the simulation. These air parcels persist and can be followed throughout the simulation. During the runtime of the host-model (Eulerian grid point CCM), the Lagrangian transport model performs online calculations to advance these air parcels using the host model's dynamics. The model output is then used to track back air parcels from the AMA to the PBL. A brief description of the CCM with Lagrangian transport will be presented after the description of the trajectory model.

### 2.1   Trajectory model data

The trajectory model, which was used to calculate the backward trajectories starting in the monsoon region, was described by Garny and Randel (2016). As for the kinematic calculations presented by Garny and Randel (2016) we have used a time step of 0.5 h and input data from six-hourly ERA-Interim data (Dee et al., 2011) with a horizontal grid spacing of $1.5° \times 1.5°$ on 37 pressure levels from 1000 hPa (surface) to 1 hPa to calculate the trajectories.

For our trajectory calculations each day during Northern Hemisphere (NH) summer (01 June to 31 August) a set of trajectories with one degree horizontal grid spacing in the region 10-50° N $\times$ 0-150° E at 150 hPa was initialized at 00 UTC. This period covers the late ramp-up and the mature phase of the AMA (Mason and Anderson, 1963). The trajectories were calculated backward up to 90 days. Output (e.g. trajectory position and surface pressure below the trajectory) was produced every six hours and all analyses for the trajectory model data described here were performed offline on the output data. In total 14 summer monsoon seasons in the period 1979-2013 have been selected as they showed a rather eastward or westward displacement of the AMA. Therefore, a modified version of the so-called South Asian High Index (SAHI; Wei et al., 2014), which measures the longitudinal displacement of the AMA, has been employed. A detailed explanation for the choice of the years and a description of the selection process is given in Sect. 2.3 and the Appendix A3. In the following, results from the trajectory model will be also indicated via the abbreviation TRJ (short for TRaJectory).





We note here that there is a variety of approaches to calculate trajectories from or to the upper troposphere in the AMA region. For example, Bergman et al. (2013) mainly focused on kinematic trajectories to investigate PBL to AMA transport. Similarly, Fan et al. (2017) used kinematic trajectories to calculate the transport from the PBL to the UT in the AMA region. Other studies employed kinematic and/or diabatic trajectories in combination with observed cloud top heights to investigate transport processes in the ASM region (e.g. Bucci et al., 2020; Legras and Bucci, 2020) or hybrid diabatic trajectories (e.g. Vogel et al., 2015, 2019). Based on Lagrangian transport model data from the CCM, we will also address the influence of (hybrid) diabatic versus (hybrid) kinematic trajectories.

## 2.2 EMAC-ATTILA data

In this study, we also exploit Lagrangian model data from a CCM simulation described by Brinkop and Jöckel (2019). In this simulation, the CCM EMAC (ECHAM/MESSy Atmospheric Chemistry; Jöckel et al., 2016), was run together with the most recent version of the submodel ATTILA (Atmospheric Tracer Transport In a LAgrangian model; Reithmeier and Sausen, 2002), which calculates the Lagrangian transport of air parcels (Brinkop and Jöckel, 2019). Within this EMAC-ATTILA simulation about 1.16 million air parcels, which represent the global atmosphere, are initialized at the beginning of the simulation and are consequently transported according to the CCM's wind fields (Brinkop and Jöckel, 2019). ATTILA can be operated with diabatic and kinematic transport as provided by the CCM (Brinkop and Jöckel, 2019). Further, since its newest update ATTILA can be used with a Lagrangian convection parametrization, which is consistent with the grid-point convection scheme: based on the mass fluxes of the convection scheme - as provided by the host model - air parcels within a column have a probability to be vertically displaced due to convection such that there is no net vertical air parcel transport between grid boxes, i.e. the number of air parcels in each grid box remains unchanged (Brinkop and Jöckel, 2019, see in particular their Section 2.2.4). The EMAC-ATTILA data used in this study incorporates the effects of parametrized Lagrangian convective transport and either a diabatic or kinematic vertical velocity scheme was employed. In the following, the corresponding model results will be referred to as LG-D and LG-K, respectively.

The underlying EMAC simulation has a grid point spacing of roughly $\sim 2.8° \times 2.8°$ and the model top is located roughly at 0.01 hPa (Brinkop and Jöckel, 2019). The meteorology of the grid-point model evolves freely (Brinkop and Jöckel, 2019), i.e. it is not restrained by observed meteorology, and is hence described as free-running. Here we employ the ten hourly output of the model data. For further details regarding the simulation setup see the Appendix (Sect. A1) and Brinkop and Jöckel (2019).

## 2.3 Analysis method

To analyze transport from the PBL to the AMA, we retrace the pathways of individual trajectories or air parcels for both, the trajectory model and EMAC-ATTILA. For both modelling approaches, the trajectories are followed up to 90 days backward in time. When the pressure below the trajectory is larger than 0.85 times surface pressure, we assume that the trajectory has encountered the PBL as described by Bergman et al. (2013). For our analyses the focus will lie on trajectories that start within the AMA, unless otherwise noted. We define the AMA boundary using a geopotential height anomaly criterion with respect to the 50° S-50° N mean as proposed by Barret et al. (2016; see details in the Appendix A2). For the trajectory model data



the boundary of the AMA was determined via a geopotential height anomaly (GPHA) threshold of 280 m using ERA-Interim data (see Appendix A2 for details). Consequently, all trajectories that show a GPHA of at least 280 m are said to be located within the AMA. Sensitivity studies with a GPHA of 260 m for the trajectory model data showed that our qualitative results are not overly sensitive to the choice of the GPHA threshold. For the EMAC-ATTILA analyses (Sect. 4) a separate threshold
(of 295 m) for the boundary of the AMA was determined (see Sect. A2). This was necessary as the EMAC-ATTILA simulation is free-running (as noted before) and thus develops slightly different climatological states e.g. with respect to the temperature (Jöckel et al., 2016). As the number of trajectories that start within the AMA varies from year to year in our analyses, we first calculate the respective distributions before producing the multiannual mean. Hence, each year contributes equally to the presented analyses.

### 2.3.1 TRJ

For the trajectory model, the daily initialized (backward) trajectories are followed backwards in time based on the six hourly output of the data. In total, trajectory model data for 14 NH summer seasons (from 01 June to 31 August) out of the period 1979 to 2013 have been analyzed. Choosing these 14 years was motivated by the finding that anomalies of the vertical velocity in the AMA region are related to the position of the AMA (Nützel et al., 2016, their Fig. 14). Hence, we assumed that the
transport properties might be related to the mean position of the AMA. Accordingly, the selected 14 summer seasons have been chosen out of the period 1979 to 2013, as the anticyclone showed a rather eastward (seven summer seasons) or westward location (seven summer seasons) during these years. Further details on the selection method are given in the Appendix A3.
In Fig. 1 we show the differences of vertical velocities at 150 hPa for the two composites (west minus east). Stronger upward motion over the Indian subcontinent and the Tibetan Plateau (TP) is found for the west composite compared to east composite,
whereas the years with an eastward shifted AMA show stronger upward motion to the east. Here, we note that we will focus on the joint analyses of all 14 NH summer seasons for the majority of our analyses and address differences between east and west years with additional dedicated analyses.

### 2.3.2 EMAC-ATTILA

For the EMAC-ATTILA simulation we use the ten hourly output of the model data and perform our analyses for 30 NH summer
seasons (again, 01 June – 31 August) from 1981 to 2010. Due to a processing error for the LG-K data the year 2008 had to be removed. As already described before, in EMAC-ATTILA the trajectories persist throughout the full simulation period, hence, the individual air parcels are distributed freely. Thus, we perform our analyses at each output time step (every ten hours) for all parcels that are located inside the AMA, i.e. satisfy the geopotential height anomaly criterion and are located on the NH within 60° W-180° E, within the pressure range of 140–160 hPa. As mentioned before, for the free-running EMAC-ATTILA
simulation a different geopotential height anomaly threshold needed to be derived than for the TRJ data (see Sect. A2).



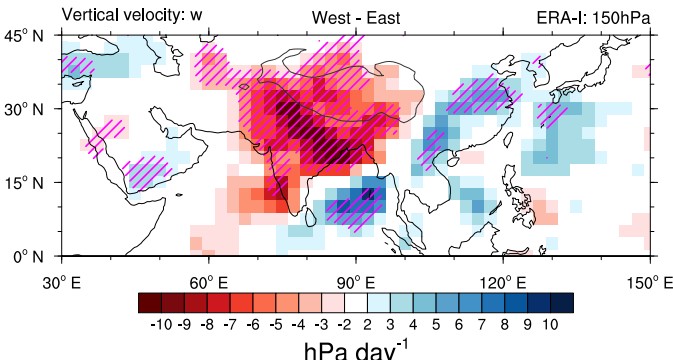

**Figure 1.** Composite difference of ERA-Interim vertical velocities (in hPa day$^{-1}$) at 150 hPa (west minus east). Magenta hatching indicates the significance level of 10%. The outlines of the TP are given as black contours based on ERA-Interim data. The vertical wind fields were horizontally smoothed prior to the analysis.

## 2.4 Reanalysis data

ERA-Interim data (Dee et al., 2011; European Centre for Medium-Range Weather Forecast (ECMWF), 2011) at $1.5° \times 1.5°$ horizontal grid spacing are used to calculate the TRJ data. Additionally, ERA-Interim data (partly also at different resolutions) are employed for the interpretation of the TRJ data (e.g. to provide corresponding meteorological fields, land–sea masks, orog-
raphy etc.) and in complementing analyses.

## 3 Trajectory model results

As already stated, we will focus on the analysis of the trajectory model results (TRJ). Figure 2 shows the starting probabilities of trajectories located within the AMA, i.e. the fraction of days during JJA for which the starting positions of the trajectories
are located within the AMA at 150 hPa at a certain grid point for the trajectory model calculations. The corresponding starting probabilities for years with a rather eastward or westward displacement of the AMA (see Appendix A3) are given as cyan solid and purple dashed contours, respectively.

### 3.1 Climatology and interannual variability

First, we will start investigating the climatological properties of the transport pathways and the PBL sources of air masses from
the AMA in the TRJ data with additional notes on the interannual variability. The intraseasonal variability will be discussed thereafter.

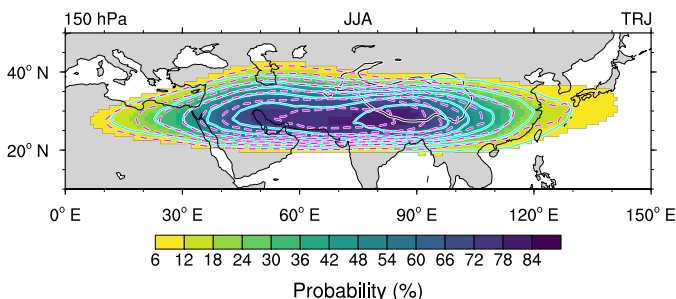

**Figure 2.** Probabilities (%) of starting locations for trajectories that start within the AMA at 150 hPa during JJA in the TRJ calculations. Trajectories have been started at each $1° \times 1°$ point per day within the region 10-50° N × 0-150° E and are said to be located within the AMA if the geopotential height anomaly from ERA-Interim (at 1.5° grid spacing) was higher than 280 m (see text for further details). Black contours show the 3 km outline of ERA-Interim orography, highlighting the TP. Purple dashed (cyan solid) contours (starting at 12% in steps of 12%) show the starting probabilities for the west (east) composites (see Sect. 2.3 for details).

### 3.1.1 Transport pathways

Figure 3 shows the probability density of crossing locations of trajectories for specific height levels, i.e. 200 hPa, 300 hPa, 400 hPa and the boundary layer (defined as 0.85 times surface pressure) in the TRJ calculations. This analysis is analogous to the analysis shown e.g. in Fig. 4 of Bergman et al. (2013). In all panels only trajectories that reach the PBL within 90 days of their release are accounted for. Our results show that during JJA on a climatological basis, AMA air mass sources come from a broad region in the PBL in Asia (bottom right panel) and with increasing height, the upward transport of air masses focuses on the eastern side of (or below) the AMA. Thus, our multiannual trajectory analyses support the findings for August 2011 presented by Bergman et al. (2013) with respect to the final crossing points of PBL to AMA trajectories.

However, we point out that by construction this analysis only captures the regions of upward transport to the AMA and not necessarily the full three-dimensional pathways. To highlight this difference, Fig. 4 shows the density of trajectories that have fallen below 200 hPa and have risen again above 195 hPa (backward in time). This analysis points out the locations of downward transport. Approximately half of all PBL crossing trajectories experience this re-circulation at this pressure level in the depicted region.

To simplify the interpretation a clarifying schematic for two hypothetical PBL-crossing trajectories (trj1 and trj2) is shown in Fig. 4 (right panel): The positions of trj1 and trj2 at the red dots would be noted in Fig. 3 - showing regions of upward transport, i.e. the final crossing points of a certain level of the trajectories. Whereas the position of trj1 at the blue dot would be noted in Fig. 4 - highlighting regions of downward transport.

Thus, the emerging picture is in agreement with upward circling, which follows the first updraft as described by Vogel et al. (2019) and Legras and Bucci (2020). The transport pathways further fit with the distribution of mean vertical velocities in the UTLS in the monsoon region (e.g. Nützel et al., 2016, their Fig. 10) as well as tracer transport and distribution as discussed by Pan et al. (2016; cf. also their discussion on the large scale circulation in the AMA region).



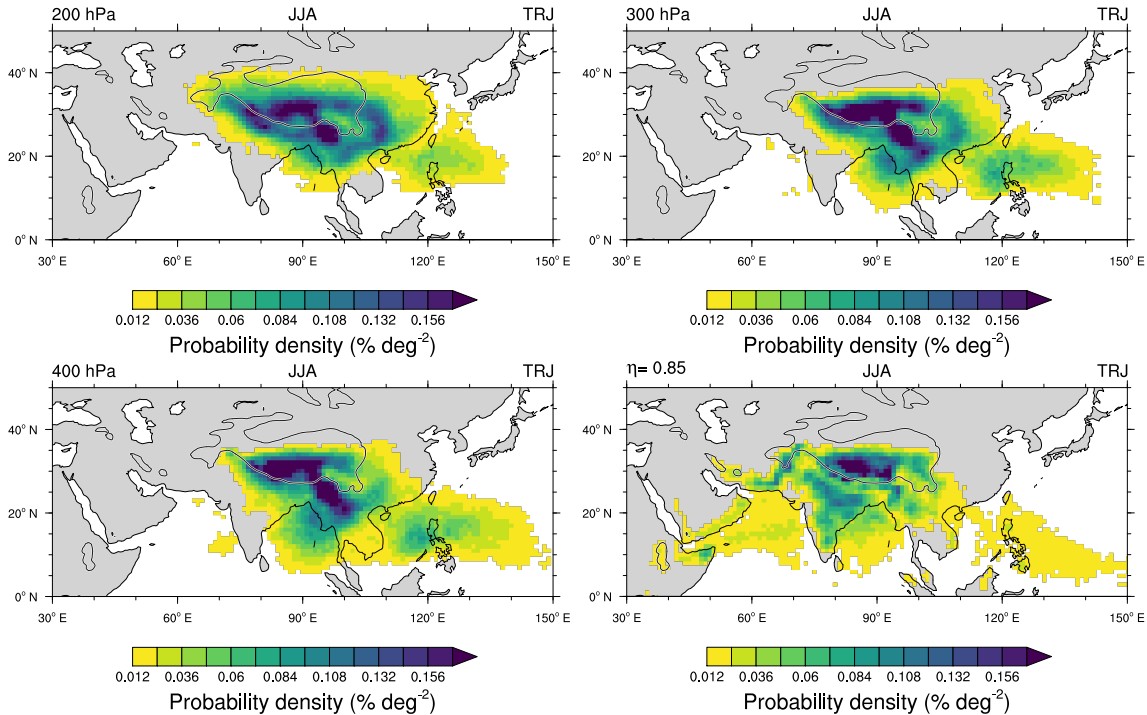

**Figure 3.** Probability density (% deg$^{-2}$) of trajectory (upward) crossings at (top left) 200 hPa, (top right) 300 hPa, (bottom left) 400 hPa and (bottom right) the PBL (defined as 0.85 times surface pressure) for trajectories that start within the AMA and cross the PBL (as defined before). As noted before, for the 14 years the individual distributions have been calculated and averaged afterwards, i.e. each year contributes equally to the probability density (also for subsequent analyses). Here and in the following plots, if the last bin of the colour bar is denoted by a triangle, it contains all values up to the maximum of the field, which is plotted.

To get a better picture of the full transport pathways, we show the distributions of PBL crossing trajectories as a longitude vs. log-pressure height cross section in Fig. 5. The scale height was chosen as 7 km as was done e.g. by Abalos et al. (2017)

225 and the reference pressure of 1013.25 hPa as in the base model of the EMAC-ATTILA simulations (cf. Roeckner et al., 2003, for details on ECHAM5). The individual panels show the temporal evolution of the trajectories that start within the AMA, 1 day, 2.5 days, 5 days and 15 days prior to their release (top left to bottom right panel, respectively). For orientation purposes meteorological data from ERA-Interim is overlaid (see Fig. caption for details).

Obviously, as noted by Bergman et al. (2013) the main upward transport occurs on the eastern side below the anticyclone

230 (centered around ∼90° E), however, as already indicated above the trajectories start to fill the AMA well below the initial release height (150 hPa). Again this points towards an upward circling already considerably below 150 hPa for some of the trajectories and refines the original conduit schematic as depicted and discussed by Bergman et al. (2013). As pointed out before, this is also in accordance with the large scale uplift above 360 K described by Vogel et al. (2019) in particular, if one considers that the release height of our trajectories is mostly above 360 K (see cyan lines in Fig. 5). It is worth noting that 15





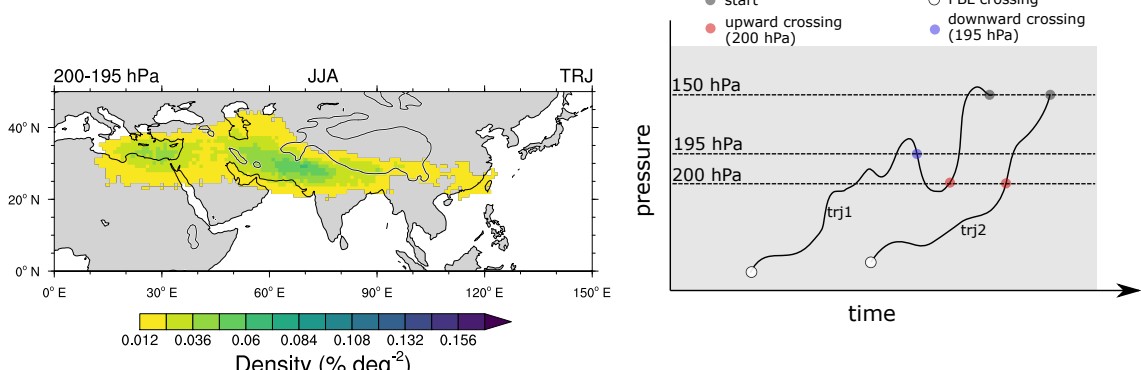

**Figure 4.** (Left) Density (% deg$^{-2}$) of trajectory (downward) crossings at 195 hPa for trajectories that start within the AMA and cross the PBL (as defined before) and fall below 200 hPa before they reach the final destination at 150 hPa. (Right) Schematic of trajectory crossings described in the left panel of this figure and in Fig. 3. See text for details.

days prior to release a considerable fraction of trajectories has reached the PBL above the TP (maximum in the density around 5 km and 70-100° E in Fig. 5 lower right panel).

The complementing latitude versus log-pressure height cross section of the climatological trajectory positions for JJA is shown in Fig. 6. Here, the trajectory positions (left) 5 and (right) 15 days prior to their arrival at 150 hPa are depicted. Again, meteorological data from ERA-Interim is overlaid to facilitate the interpretation. The trajectory distribution around the AMA height levels is tilted from North to South, in agreement with a tilt of the isentropic levels (see cyan lines in Fig. 6). We note that the distribution shows high values above or around the slopes of the Himalayan mountains (roughly at 30° N) and that over time more and more trajectories reach their PBL source region over the TP (max. around 5 km and 30-35° N) and to its south.

We will now address the sensitivity of the presented results with respect to east-west shifts of the AMA on interannual time scales. Therefore, Fig. 7 shows the differences in the upward transport regions for west minus east years. Differences are clear in the upper level (200 hPa) and fit to the differences in the vertical wind fields in the UT (cf. 150 hPa level in Fig. 1). The differences are less pronounced at the top of the PBL (defined as 0.85 times surface pressure).

To capture the differences in the full pathways, Fig. 8 shows the differences of the density distributions of the trajectories (west minus east years) as longitude vs. pressure cross section on individual dates with respect to the initialization date. Whereas differences are pronounced and significant shortly after the release of the trajectories in the UT, they get less pronounced and clearly less significant at lower levels. Overall, we note that the qualitative results regarding the transport pathways remain stable.



**Figure 5.** Longitude versus log-pressure height cross sections of density distributions (% deg$^{-1}$ km$^{-1}$) of trajectory positions for PBL crossing trajectories (top left) 1 day, (top right) 2.5 days, (bottom left) 5 days and (bottom right) 15 days prior to their arrival at 150 hPa within the AMA. The three-dimensional probabilities were integrated over 0-50° N. Please note the different colour bars. The 150 hPa level corresponds roughly to 13.5 km and the 200 hPa level is located at roughly 11.5 km in the plot. For this analysis, once trajectories reach the PBL they are not further tracked and will be noted at the first PBL-crossing point also later in time. Please note - also for upcoming figures - that the maximum in the distribution at 4-6 km, e.g. present in the bottom right panel, is related to the TP. Cyan lines indicate potential temperature levels at 30° N starting at 340 K to 380 K in steps of 10 K and 20 K afterwards (380 K to 480 K). Black contours indicate meridional winds at 30° N in steps of 3 m s$^{-1}$. Negative, i.e. southward, winds are dashed and the zero wind line is given in orange. Meteorological data based on ERA-Interim is flagged out below the grey line, which indicates the ERA-Interim minimum surface pressure in the region 0-50° N of the time average JJA for the trajectory years.

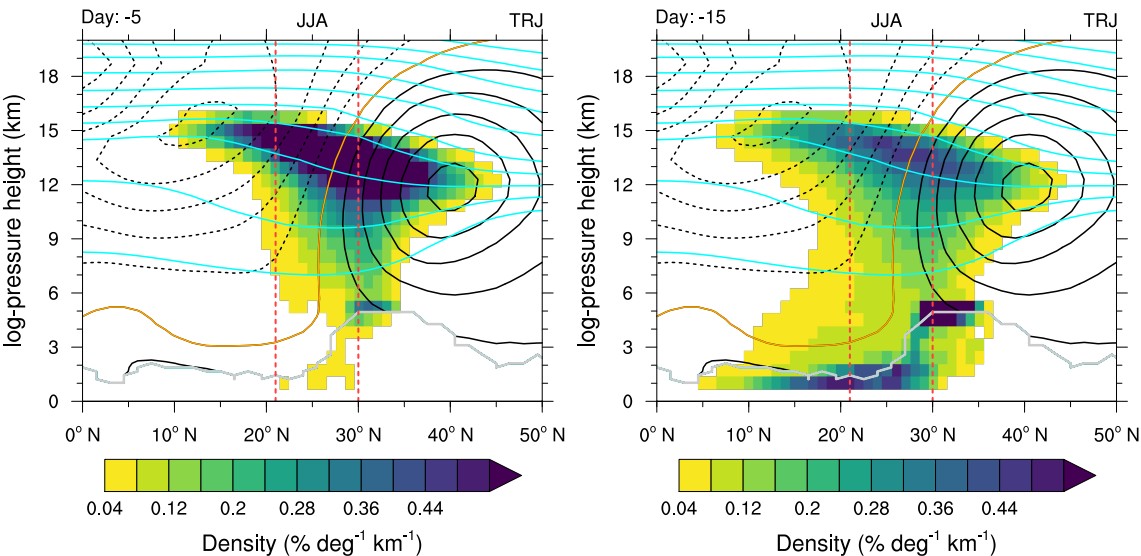

**Figure 6.** Latitude versus log-pressure height cross sections of density distributions (% deg$^{-1}$ km$^{-1}$) of trajectory positions for PBL crossing trajectories 5 days and 15 days prior to their arrival at 150 hPa within the AMA. The three-dimensional probabilities were integrated over 60-140° E. The 150 hPa level corresponds roughly to 13.5 km and the 200 hPa level is located at roughly 11.5 km in the plot. For this analysis, once trajectories reach the PBL they are not further tracked and will be noted at the first PBL-crossing point also later in time. Cyan lines indicate potential temperature levels averaged over 0-120° E starting at 340 K to 380 K in steps of 10 K and 20 K afterwards (380 to 480 K). Black contours indicate zonal winds averaged over 0-120° E in steps of 5 m s$^{-1}$. Negative, i.e. westward, winds are dashed and the zero wind line is given in orange. Meteorological data based on ERA-Interim is flagged out below the grey line, which indicates the ERA-Interim minimum surface pressure in the region 0-120° E of the time average JJA for the trajectory years.

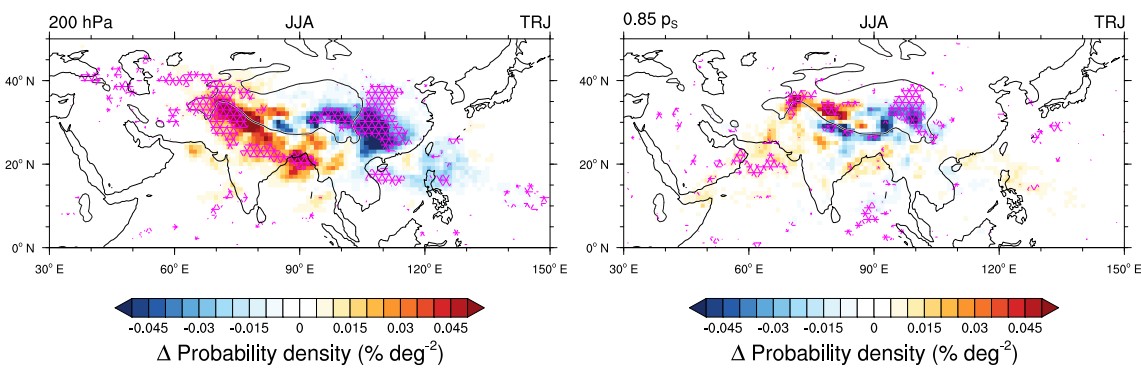

**Figure 7.** Difference (west minus east years) of probability densities (% deg$^{-2}$) of trajectory (upward) crossings at 200 hPa and the PBL (defined as 0.85 times surface pressure) for trajectories that start within the AMA and cross the PBL (as defined before). The underlying fields have been horizontally smoothed and the significance level of 0.1 is noted via magenta hatching.

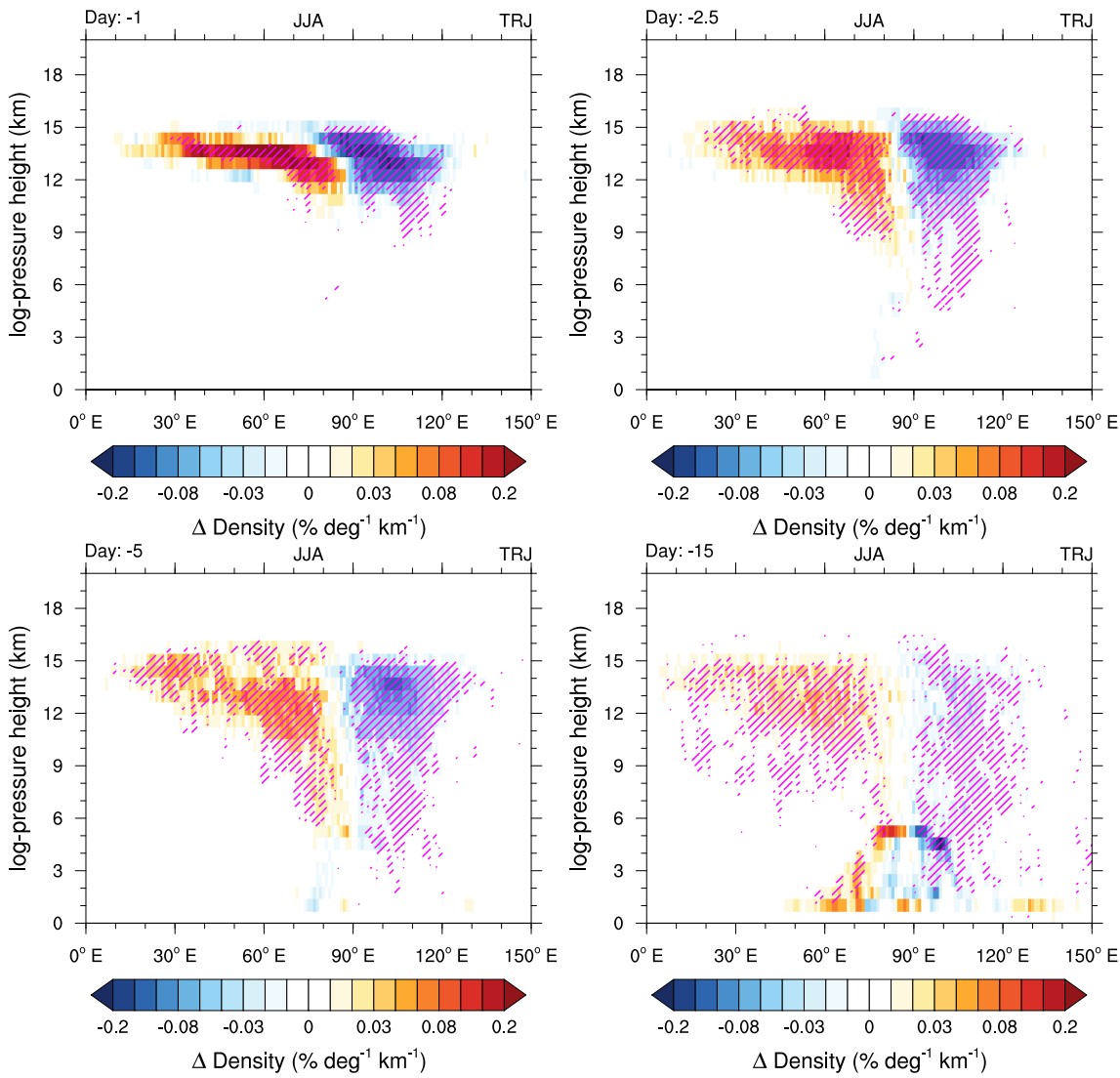

**Figure 8.** Longitude versus log-pressure height cross sections of the difference (west minus east years) of the density distributions (% deg$^{-1}$ km$^{-1}$) of trajectory positions for PBL crossing trajectories 1 day, 2.5 days, 5 days and 15 days prior to their arrival at 150 hPa within the AMA. The three-dimensional probabilities were integrated over 0-50° N. The 150 hPa level corresponds roughly to 13.5 km and the 200 hPa level is located at roughly 11.5 km in the plot. Once trajectories reach the PBL they are not further transported and will be noted at the crossing point also later in time.





### 3.1.2 Boundary layer source regions

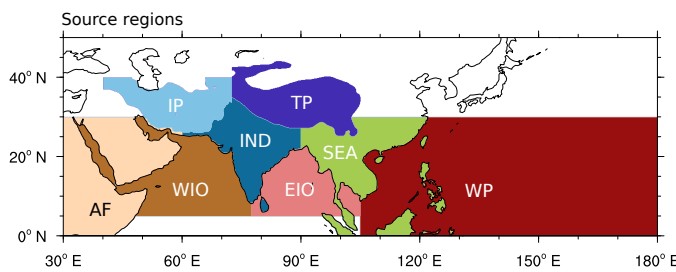

**Figure 9.** Source regions based on ERA-Interim orography and land-sea mask data at $0.125°\times0.125°$ grid spacing for the TRJ calculations. See text for details.

In the following, we want to further analyze from which PBL source regions air masses within the AMA originate. Therefore, Fig. 9 shows the definition of the individual source regions. The TP (mainly the Tibetan Plateau) and IP (mainly the so-called Iranian Plateau) regions are defined as regions with a surface elevation of more than 2 km and 0.5 km in the boxes 75–110° E × 25–45° N and 40–75° E × 25–40° N, respectively. The other source regions are named AF (mainly parts of Africa and the Arabian Peninsula), WIO (Western Indian Ocean), EIO (Eastern Indian Ocean), IND (mainly the Indian subcontinent), SEA (mainly consisting of Southeast Asia and parts of Southeast China) and the WP (West Pacific) region.

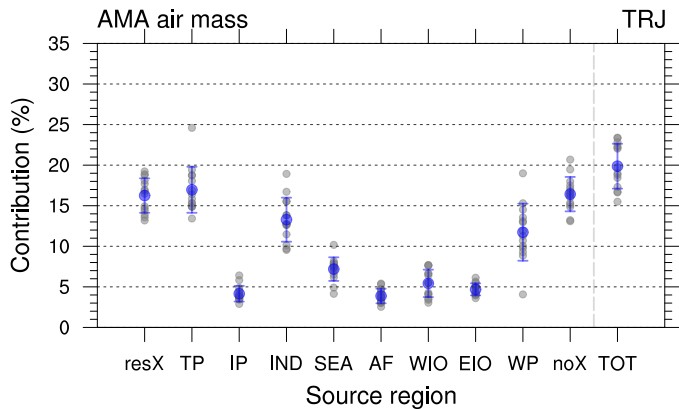

**Figure 10.** Contributions from different source regions to AMA air masses at 150 hPa during JJA. The categories resX and noX correspond to the trajectories that reached the PBL outside the defined source regions (see Fig. 9) or did not reach the PBL within 90 days prior to their start, respectively. TOT corresponds to the total numbers of trajectories released within the AMA, where 1 percent corresponds to 4000 trajectories. The mean values are given by blue dots (with blue whiskers for the interannual standard deviation), whereas individual years are shown as grey dots.

The mean contribution of individual source regions (blue dots) in the TRJ simulation and their interannual variation (translucent grey dots and whiskers) are shown in Fig. 10. The largest contributions from the named source regions are found from the





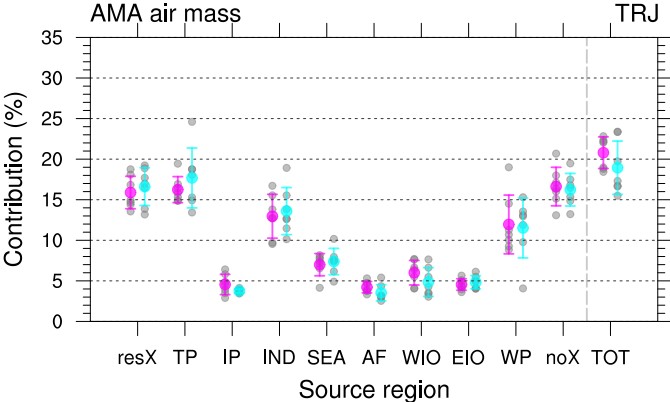

**Figure 11.** Contributions of PBL air masses to the AMA at 150 hPa split according to the east (cyan) and west (purple) location of the AMA with individual years given as grey dots. The cyan and purple whiskers mark the interannual standard deviation for the east and west composite, respectively. For TOT the total number of trajectories located within the AMA are reported, where 1 % corresponds to 4000 trajectories.

TP region (around 17%), the IND region (around 13%) and the WP region (around 12%). However, we note that the densities of PBL crossings are larger for the TP and IND region than for the WP region (see Fig. 3). There is also a considerable fraction of trajectories of around 16% that encounter the PBL outside the named source regions (resX) or do not encounter the PBL within 90 days prior to release (noX).

There is strong interannual variability regarding the sources of the AMA as indicated by relatively large whiskers and a considerable spread of the contributions in individual monsoon seasons. Nevertheless, the aforementioned regions, namely TP, IND and WP, are more important for the AMA composition in the TRJ simulation in almost all years than the other source regions. The intraseasonal variability of these source regions will be discussed along with the variability of the transport pathways in the next section (Sect. 3.2).

Figure 11, shows the contributions of different source regions to the AMA split according to the rather eastward (cyan) or westward (purple) location of the AMA. This analysis shows that there are no systematic differences in the mean source region contributions according to the east–west location of the AMA on interannual timescales and in particular the large interannual variability renders the slight differences between the two composites insignificant. This is in agreement with the previous statement that the main transport pathways did not change qualitatively with respect to the east–west displacement of the AMA

on an interannual basis and that the boundary layer changes are relatively small or partly compensating within the different source regions as for instance for the TP region (see Fig. 7). With respect to the interannual variability within the composites, the TP region and the total number of trajectories (TOT) show enhanced variability in the east composite whereas reduced variability is found for the IP contribution. Whether this result is robust or not, is unclear. Slightly more trajectories are located within the AMA for years in which the AMA is displaced to the west (in agreement with the higher maximum in the contour



lines for westward location of the AMA in Fig. 2). This difference, however, is not significant as there is strong interannual

variability as indicated from the interannual standard deviation (included as whiskers in Fig. 11).



## 3.2 Intraseasonal variability

### 3.2.1 Transport pathways

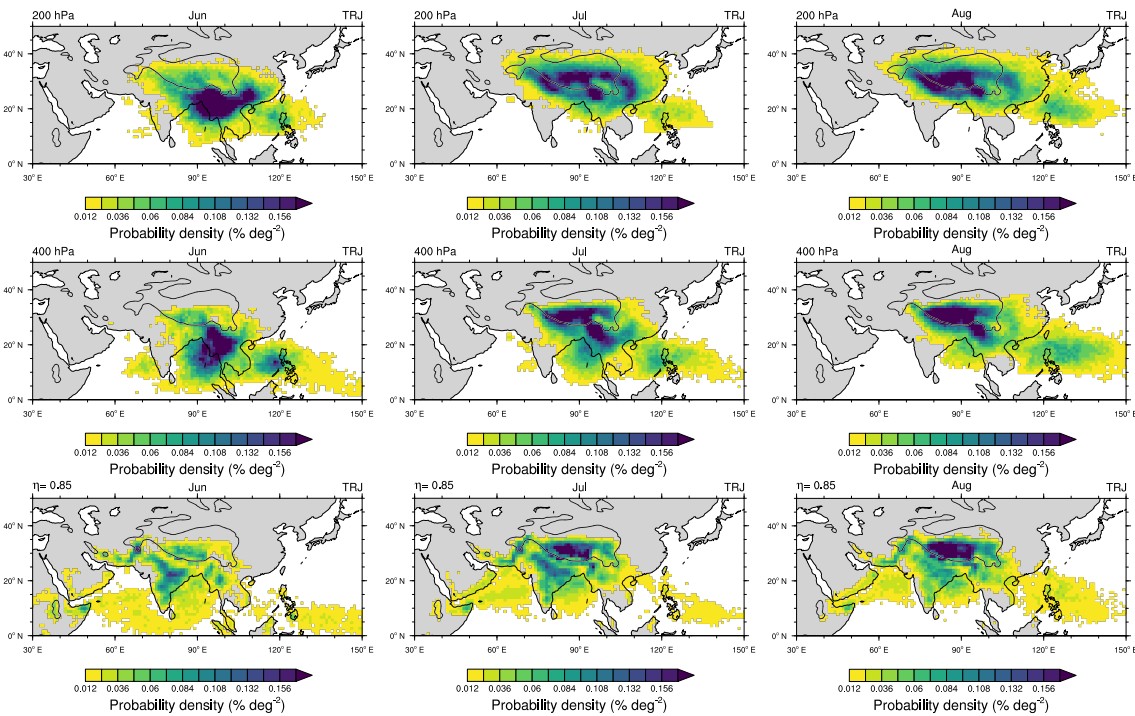

**Figure 12.** Probability density (% deg$^{-2}$) of trajectory (upward) crossings at (top row) 200 hPa, (middle row) 400 hPa and (bottom row) the PBL as in Fig. 3 but split according to (left column) June, (middle column) July and (right column) August.

To further analyze the subseasonal variability with respect to the PBL source regions and the transport pathways, Fig. 12
(analogous to Fig. 3) shows maps of final boundary layer and pressure level crossings split according to June, July and August, respectively. As can be seen from these plots the PBL crossings shift over continental Asia over the course of the monsoon season from June to August. Furthermore, the regions of upward transport, which are mainly centered over the eastern Indian Ocean (Bay of Bengal) and adjacent continental regions at 200 and 400 hPa in June, shift northwards towards the TP in July and August.

A more quantitative view of this northward shift is presented in Fig. 13, which show the distributions of the latitudinal position of PBL crossings for June (blue), July (red) and August (purple) of trajectories starting in the AMA. In particular, the modal value in June at 5° N is clearly reduced in July (and August) and the contributions around 30° N roughly double from June to July. The interannual variability depicted as dashed lines in Fig. 13, allows to draw the conclusion that this is a typical behaviour throughout the monsoon season.





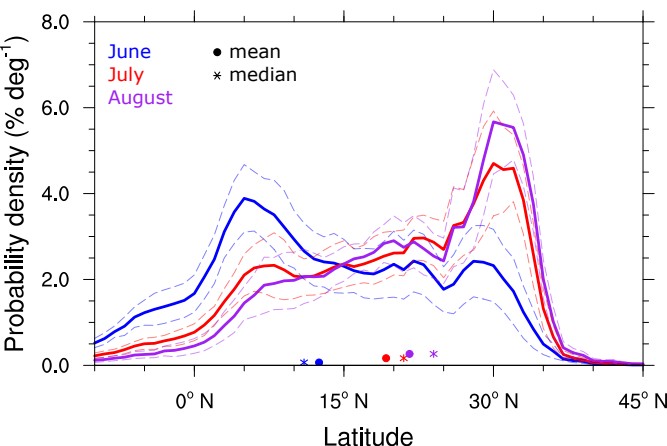

**Figure 13.** Probability density (% deg$^{-1}$) with respect to latitude of trajectory intersections with the PBL split according to June (blue), July (red) and August (purple). Mean (dots) and median (crosses) are given as well. Dashed lines mark the interannual standard deviation.

For a complementing view of the transport pathways during June to August, Fig. 14 shows the distributions of the trajectories in a latitude versus log-pressure height cross section 5 and 15 days before the trajectories encounter their starting position at 150 hPa. It is shown that the trajectory locations shift from south to north during the evolution of the ASM from June to August. In August, the AMA is located above the TP and transport from the TP into the AMA occurs vertically. We emphasize the clear shift of the maximum density at about 6 km to 10 km from approximately 20° N in June to 30° N in August.

This northward shift of the PBL source regions and the transport pathways is consistent with the northward shift of the region of low outgoing longwave radiation and the AMA (Nützel et al., 2016, their Fig. 12; see also the related discussion) and the monsoon (precipitation) itself (e.g. Wang and LinHo, 2002; Yihui and Chan, 2005). This northward propagation can also be seen in deep convective activity as monitored by satellite measurements, where deep convection (up to 150 hPa) over the TP is rare in June and becomes more prominent in July and August (Devasthale and Fueglistaler, 2010).



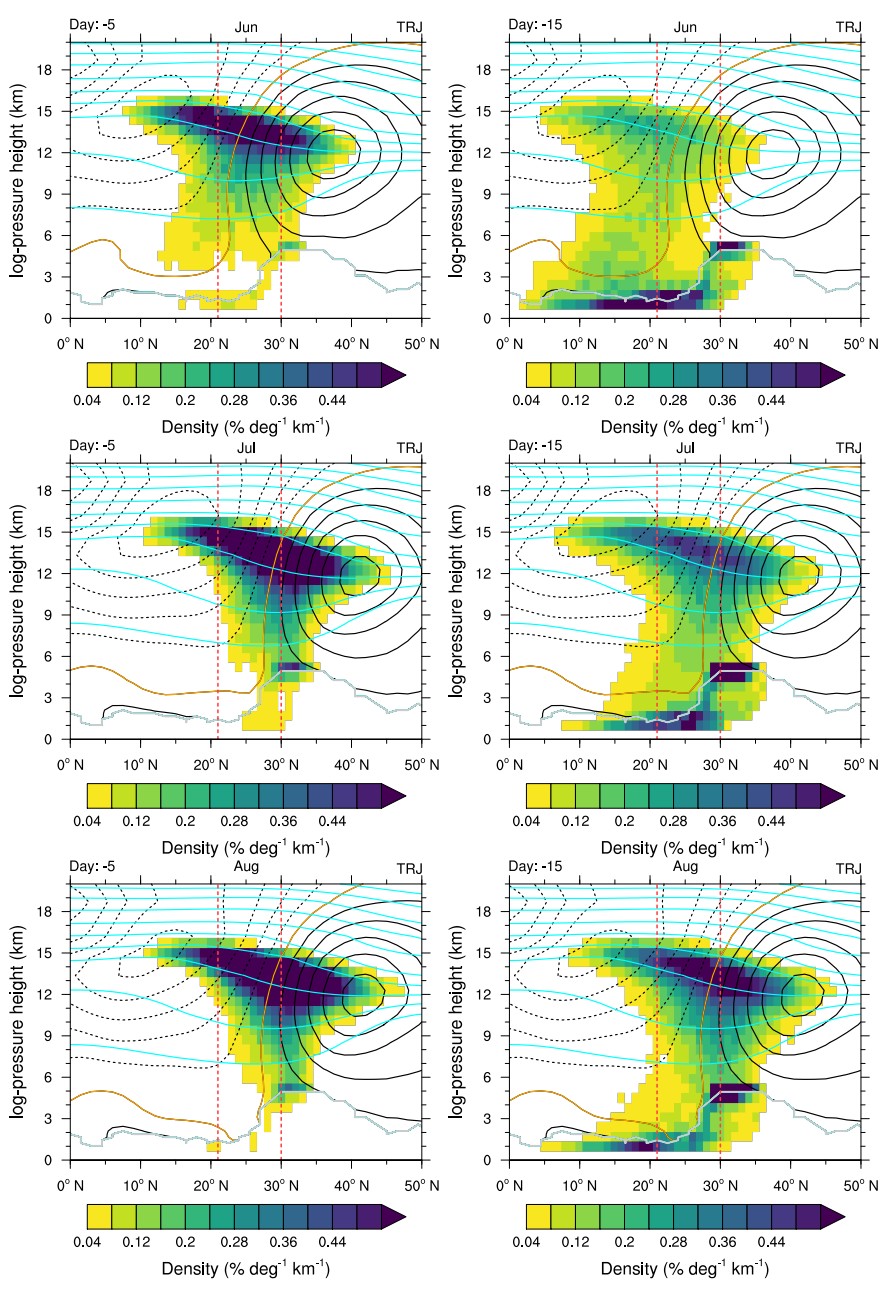

**Figure 14.** As in Fig. 6 for (left column) 5 and (right column) 15 days prior to their final position at 150 hPa, split according to (top row) June, (middle row) July and (bottom row) August. Again the three-dimensional probabilities were integrated over 60-140° E. For orientation purposes red vertical dashed lines at 21° N and 30° N, roughly indicate the maxima in the distributions between 6 and 12 km for June and August for the trajectories 15 day prior to their arrival at 150 hPa, respectively. Meteorological data from ERA-Interim is presented as in Fig. 6 but separated for June, July and August, respectively.



### 3.2.2 Boundary layer sources

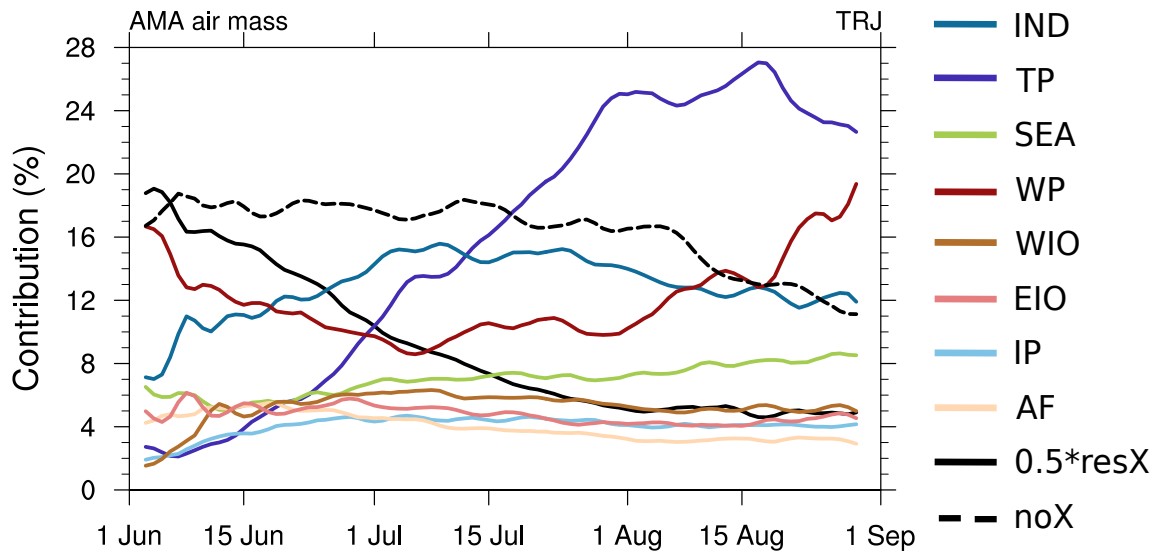

**Figure 15.** Temporal evolution of source region contribution to the AMA air masses at 150 hPa in the TRJ calculation. To fit the scale the resX category was scaled by 0.5. All contributions have been smoothed via 5 day running means (weights of $[\frac{1}{9}, \frac{2}{9}, \frac{3}{9}, \frac{2}{9}, \frac{1}{9}]$).

Besides the strong interannual variability the AMA is also known for its intraseasonal/subseasonal variability (see e.g. Fig. 5 in Garny and Randel, 2013, showing both interannual and intraseasonal variability). Hence, we now concentrate on the contribution of individual PBL source regions to the AMA air masses with particular focus on the subseasonal variability. Fig. 15 shows the temporal evolution of the source region contributions in the TRJ simulation. The most prominent change is the

increase of the TP contribution from below 4% in early June to more than 24% for most of August. Also, it is obvious that the fraction of non-crossing (noX) trajectories clearly decreases over time. This implies that over the monsoon season the fraction of air masses within the AMA that have recently (within the last 90 days) come from the PBL increases. Further, over the course of the monsoon season, the contributions of trajectories that cross the PBL outside the monsoon region (resX) declines noticeably. This indicates that the PBL sources focus more toward the Asian monsoon region and is in accordance with the

impression from Fig. 12. The WP region shows a minimum contribution at the beginning of July (below 10%) whereas the contributions in early June (around 16%) and end of August (around 20%) are clearly higher. For the IND region, the evolution is reversed with a peak contribution in July (~16%) and lower contributions in early June and end of August (about 8% and 12%, respectively). Apart from a small dip in early June, the contribution of the SEA region increases steadily from around 5% in mid June to approximately 9% end of August. For the AF region this behaviour seems to be reversed (from around 5% to

3%). All other source regions (WIO, EIO and IP) show some variation in June but have relatively stable contributions (between about 4-6%) during July and August.





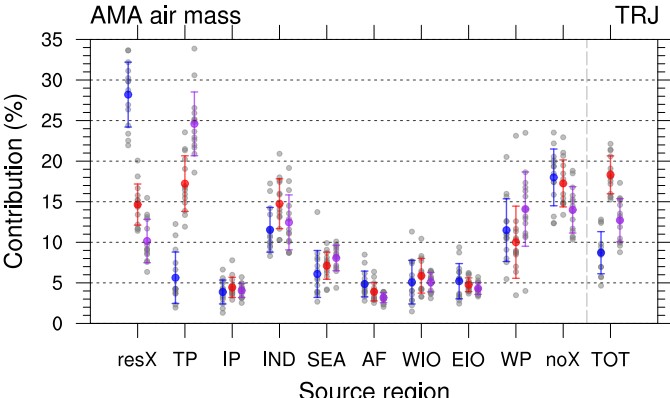

**Figure 16.** Contributions of PBL sources to the AMA at 150 hPa for the TRJ simulation over 14 years split according to June (blue), July (red) and August (purple). The interannual standard deviation is given as whiskers and the individual years are included as grey dots. For TOT 1% corresponds to 2000 trajectories.

Figure 16 shows the PBL source region contributions split according to June, July and August for the multiannual mean (coloured dots) and individual years (grey dots). This figure is closely related to the analysis shown in Fig. 15 but additionally

captures the interannual variability of the subseasonal development and allows to assess, how robust these subseasonal features are. As an example, the increase in the contribution of the TP from June to August is clearly visible and the separation of the data points for June and August indicates that this is a typical feature of the AMA air mass contributions. Indeed, the increase of the contribution of TP air masses to the AMA from June to August is present in every single year. Further, except for one year, the TP is the most important source region for air masses within the AMA in August in our analysis. Also, as the resX

contribution significantly declines from June to July/August, it is shown that the PBL source regions focus more on the ASM region, which is in accordance with Fig. 12. Further, we also note that more trajectories are located within the AMA in July than in June and August. This is in agreement with the seasonal cycle of the AMA (e.g. Garny and Randel, 2013; Nützel et al., 2016, Figs. 5 and 12, respectively) as already described by Mason and Anderson (1963). For the other source regions, the intraseasonal variations are overruled by the strong interannual variability and more years would be needed to carve out robust

differences.





## 4 EMAC-ATTILA results: a complementary view

To corroborate our results and to point out sensitivities and uncertainties, we show also the results of free-running Lagrangian CCM simulations. As already noted in Sect. 1, the Lagrangian data from these simulations can provide a complementary view because the modelling approach differs largely from the reanalysis driven trajectory data presented in Sect. 3. The EMAC-
ATTILA data contain the effect of parametrized convection and stem from two free-running simulations, in which the vertical velocity is described either by a kinematic (LG-K) or a diabatic (LG-D) scheme (cf. Brinkop and Jöckel, 2019). To be precise, the simulations feature a sigma-pressure or sigma-theta vertical coordinate (see Brinkop and Jöckel, 2019). This hybrid coordinate allows to overcome some of the problems associated with diabatic trajectories in the troposphere mentioned by Bergman et al. (2013) and by Honomichl and Pan (2020). For the analyses the forward trajectories of the EMAC-ATTILA data
were traced back. Further, all analyses were conducted based on the underlying EMAC model grid. In particular, the respective boundary layer source regions were defined based on the underlying horizontal resolution of the base model.

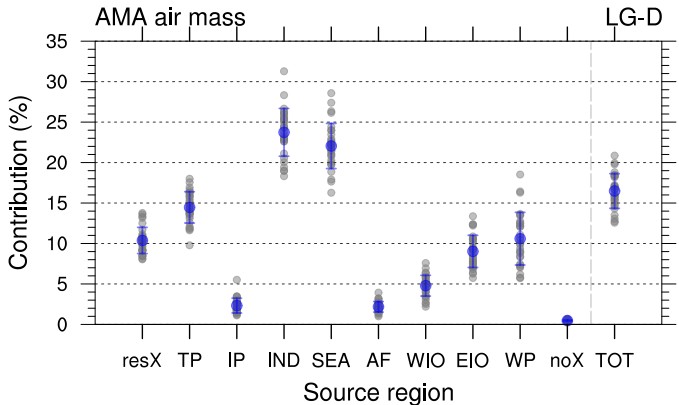

**Figure 17.** Contributions of PBL sources to the AMA around 150 hPa for the LG-D simulation. Whiskers denote the interannual standard deviation, whereas individual years are indicated via grey dots. For TOT 1% corresponds to 8000 trajectories.

First, we want to focus on features, where the LG-D simulation support the results of the TRJ calculations. Secondly, we show which results differ and where (a parametrization of) Lagrangian convection might be of importance. Finally, we also address the impact of the vertical velocity scheme by comparing the model results of the LG-D and LG-K.
We have found that the pathways of the LG-D data (see supplemental Fig. B2) look similar to the pathways shown in Fig. 5. Moreover, the LG-D data also show strong interannual variability in the source region contributions (cf. Fig. 17).

Further commonalities in the TRJ and LG-D model data results can be seen, when it comes to the evolution of PBL contributions to the AMA air masses. Both model data show an increase of the TP contribution from June to August (Figs. 15 and 18). Also, the qualitative evolution of the contribution of the WP and SEA regions – minimum contribution during July for WP
and slight increase over the monsoon period for SEA – are similar in the two model data sets.





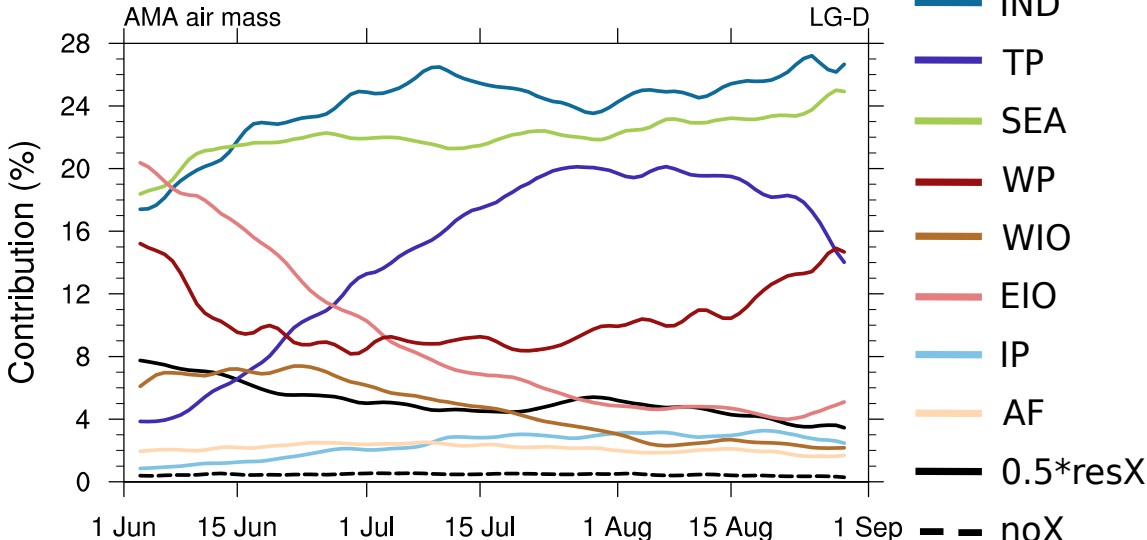

**Figure 18.** Source evolution in the LG-D data. resX data has been scaled by 0.5. All contributions have been temporally smoothed via 5 day running means (weights of $[\frac{1}{9}, \frac{2}{9}, \frac{3}{9}, \frac{2}{9}, \frac{1}{9}]$), while daily data were produced from summing up the ten hourly data for each day.

However, we have to note that quantitatively, the contributions differ between the two model data sets (cf. also Fig. 17). As an example the contribution of the TP in August is not as dominant in LG-D as in TRJ. Further, around 11% of the trajectories come from a region outside the defined sources in the LG-D, which is similar to roughly 16% in the TRJ data. However, in the TRJ data this contribution drops considerably from June to August, whereas in the LG-D data the decline is more moderate.

The differences between the TRJ and EMAC-ATTILA data are likely to be also related to the faster vertical transport in the LG-D data due to the effect of parametrized convection. As an example, the air masses that do not reach the PBL within 90 days account for more than 15% in the TRJ calculation during JJA, whereas in LG-D this value is below 1%. The differences in this fraction might also be related to the quantitative differences in the contributions of IND and SEA in the TRJ and LG-D data, namely clearly higher contributions in the LG-D data than in the TRJ calculations. An intermediate region is the EIO showing
slightly higher contributions in LG-D data, which might hint towards the importance of convective transport from this region, which is located beneath the south-eastern part of the AMA. As the contributions of IP, AF and WIO are relatively small in all model data sets, this indicates that convective transport from these regions to the AMA might not be overly important.

We stress that the above results also hold qualitatively for the LG-K data. Fig. 19 shows the differences in the contribution of source regions to the AMA air masses for LG-D minus LG-K data. Major differences are that the contribution of the TP is
not as large as in the LG-D data and that the increase over the monsoon period is less pronounced (absolute values for LG-K are shown in supplemental Fig. B4). Throughout the monsoon season, the LG-D data show overall higher contributions for TP, IND and SEA compared to the LG-K data. Almost no differences are found for the contribution of the IP, whereas lower contributions are found for the other source regions.





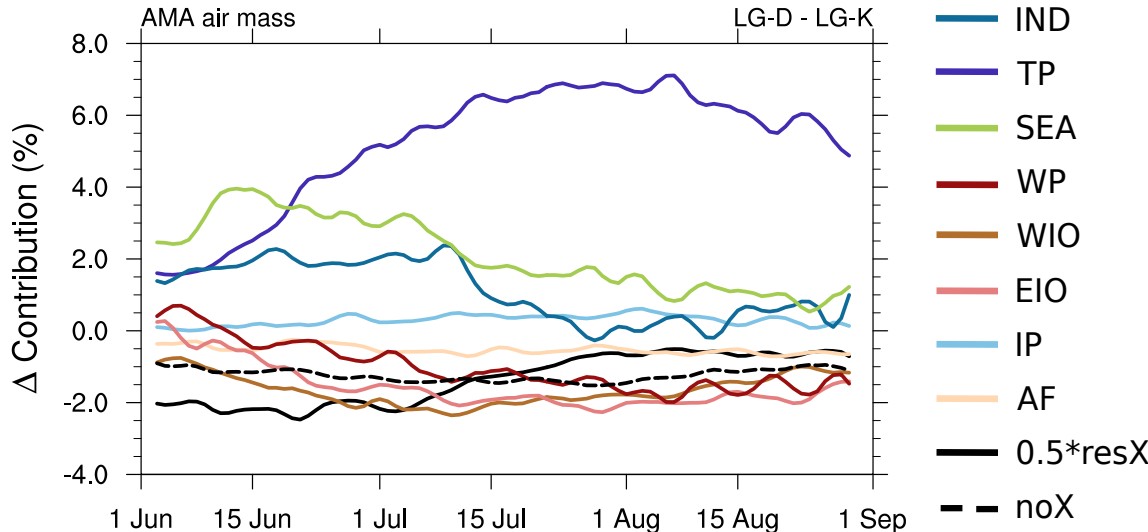

**Figure 19.** Source evolution for LG-D minus LG-K data. resX data has been scaled by 0.5. All contributions have been temporally smoothed via 5 day running means (weights of $[\frac{1}{9}, \frac{2}{9}, \frac{3}{9}, \frac{2}{9}, \frac{1}{9}]$), while daily data were produced from summing up the ten hourly data for each day. As in the LG-K data the year 2008 is missing (cf. Sect. 2.3), it was also removed in the LG-D data for this analysis.

For the LG-D data we analyzed the sensitivity of our results with respect to the method to determine the PBL. We found that

qualitatively the results do not depend on the choice of the PBL criterion, while quantitatively, the changes were in the order of switching between kinematic and diabatic trajectories (i.e. differences in the LG-D and LG-K data) while using the standard PBL criterion.

As we have found a strong increase of the TP contribution to the AMA air masses over the monsoon season in the TRJ and LG-D (less so in LG-K) data, we further analyzed for the LG-D data the change of transport properties from the TP to

the UT for June and August. Therefore, Fig. 20 shows the differences (August minus June) in the longitudinal distributions of trajectories that stem from the TP for multiple pressure levels (300-150 hPa in 50 hPa steps). In August compared to June the trajectories are more likely located in the ASM region (60-100° E), whereas in June the probability is larger east of the ASM region (and in particular the North American monsoon region sticks out). Further, also the fraction of trajectories from the TP at the different levels (June with respect to August), decreases with height (from about 90% at 300 hPa to about 70% at

150 hPa), which indicates that transport from the TP to the UT is stronger in August than in June. These results are consistent with stronger advection to the east of air masses from the TP in June compared to August due to the location of the subtropical jet.

To sum up, we want to point out that the results of EMAC-ATTILA (in particular as they come from a free-running simulation) should not be seen as validation data but rather as a help to assess which key processes are present in these data as

well. This might help to discern, which processes/source regions are not heavily dependent on the explicit representation of convection (e.g. through a parametrization) and the detailed meteorology (free-running CCM versus TRJ calculations driven





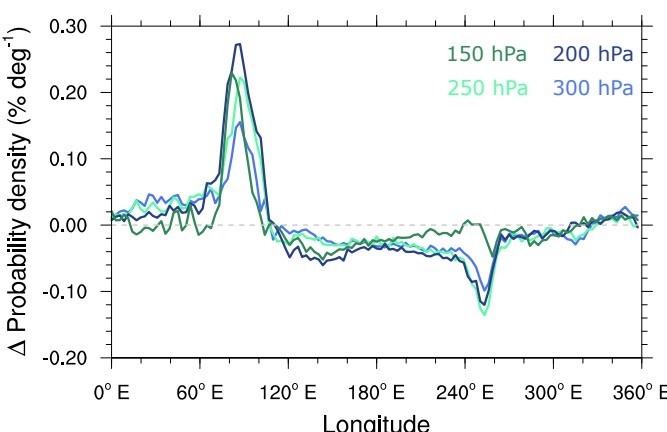

**Figure 20.** Difference (August minus June) of longitudinal probability densities of parcels that originate from the TP at various pressure levels in the UT based on 1981–2010 for the LG-D data.

by reanalysis data). As an example, the contributions from the source regions TP, WP and SEA show similar developments over the course of the monsoon period, although, the quantitative contributions partly differ. Further, the fact that the LG-D and LG-K simulations show discrepancies in parts, e.g. with respect to the mean contributions of the TP of slightly above 14% and 9% (see Fig. 17 and supplement Fig. B3), despite being driven by identical meteorological states of the host model, highlights the influence of the vertical velocity scheme to parts of the analyses. Here, we note that this might be partly already caused by the different distributions of the air parcels in LG-D vs. LG-K data: as the air parcels persist throughout the simulation and are transported with different vertical velocities, the distribution of air parcels within the AMA differs between the two model data sets (see Supplement Fig. B1), even though the same dynamical constraints are used to define the AMA. We are currently planning future work to further carve out the transport properties in the ASM region based on additional Lagrangian CCM simulations.





## 5 Discussion

### 5.1 Relation to previous modelling results and observational data

In Section 3 we have presented results regarding PBL to AMA transport based on our trajectory calculation (TRJ). We have
found that the boundary layer source distribution (Figs. 3 and 12) focuses over the ASM region (in particular over the Indian
subcontinent and the TP). Further, these distributions support previous results regarding the PBL sources of the air masses of
the AMA and its surroundings e.g. by Bergman et al. (2013) and Fan et al. (2017).

Moreover, we found similar regions of upward transport as Bergman et al. (2013), which are located on the south-eastern side
of the AMA. However, we also complemented the view about the transport pathways, i.e. the conduit proposed by Bergman
et al. (2013), by showing that air masses spread earlier in the AMA volume. In detail, we refined the picture of the vertical
conduit by showing that air masses released in the AMA circle within the AMA and fill its three-dimensional structure in
agreement with the slow uplift described by Vogel et al. (2019) and mainly experience upward (downward) transport on the
eastern (western) side of the AMA (cf. Fig. 10 presented by Nützel et al., 2016, and discussion by Pan et al., 2016). At lower
altitudes where the AMA is weak (or non-existent), vertical transport occurs mainly in (or below) the eastern part of the AMA,
in agreement with the regions of upward transport discussed by Bergman et al. (2013). The full pathways (Figs. 5 and 14) are in
agreement with the simulated tracer transport shown by Pan et al. (2016, e.g. their Figs. 2 b and c). For CO, a similar distribution
in the ASM region was found in model data by Barret et al. (2016), while in their climatological analysis of IASI satellite data
the structure was not as conclusive. Using data from the same satellite instrument but performing transient analyses, Luo et al.
(2018) came to the conclusion that this transport behaviour is also supported by the satellite data. Similarly, Vogel et al. (2019)
noted that the CO transport described by Pan et al. (2016) is in agreement with their results from a trajectory model and MIPAS
satellite data. We stress here, that the trace gas results (e.g. in modelling or satellite data) strongly depend also on the strength
and location of emissions, whereas the idealized trajectory studies simply track air mass transport.

To sum up, these results show that the transport pathways as diagnosed by (i) a trajectory model including mixing effects
(Vogel et al., 2019), (ii) a trajectory model including the effect of observed convection (Legras and Bucci, 2020), (iii) more
puristic trajectory models (Bergman et al., 2013, and this study), and (iv) forward trajectories (analyzed backwards in time)
from a Lagrangian model with parametrized convection driven by a free-running CCM (this study) are in agreement. Further,
the transport pathway is also supported by (v) analyses of CO transport within a CCM and a chemistry-transport model as
shown by Pan et al. (2016) and Barret et al. (2016) and (vi) analyses of satellite data (Luo et al., 2018; Vogel et al., 2019).
In particular, our results also show that, although there is interannual and strong intraseasonal variability, the main transport
characteristics are robust.

Regarding the source regions, our results are in agreement with some of the results found in previous studies, while keeping
in mind that there are (sometimes subtle) differences in the study design. As an example, Bergman et al. (2013) found that





roughly 27% of the all trajectories located in the AMA at 200 hPa come from the TP[1], which is similar to the mean contribution

of the TP in August in the TRJ data of this study (slightly more than 24%; about 25% for August 2011). The combined area

and contribution (again roughly 25% in August; about 26% in August 2011 in the TRJ data) of the regions IND, IP and SEA is

comparable to the area and contributions (roughly 32%)[2] of the Asian land masses excluding the TP as analyzed by Bergman

et al. (2013).

Further, Vogel et al. (2015) showed contributions of PBL sources to the AMA at 380 K. Although, the TP was not explicitly

resolved in their study, the contributions of the source regions used in their study, which cover the TP (red and green lines in

their Fig. 8) show a strong increase from June to late July. This increase is in agreement with the increase of the TP contribution

found in our study. The dependence of the TP contribution to AMA air masses on the position of the AMA is in analogy to the

relation of typhoon–AMA transport discussed by Li et al. (2017), i.e. for the TP or typhoons, entrainment of air masses uplifted

from these sources into the core of the AMA depends on the co-location of the AMA and the TP or typhoon, respectively.

Goswami et al. (1999) defined an index for the interannual Indian monsoon variability, the so-called monsoon Hadley index

(MHI), as meridional wind shear between the UT (200 hPa) and the 850 hPa level over a reference region and motivate their

definition by the relation to heating released due to precipitation in the respective region. Here we calculate the MHI from

ERA-Interim data based on JJA data. We find that the detrended MHI and (modified) SAHI are strongly anti-correlated (-

0.68) over the period 1979–2013 and in particular the anti-correlation for the years where the SAHI is anomalous (i.e. the 14

monsoon seasons for which the backward trajectories have been calculated) is even higher (-0.83). This hints that by analyzing

years with rather strong displacements of the AMA to the East or the West, we have implicitly analyzed the impact of the

detrended MHI on the transport properties from the PBL to the AMA.

## 5.2 Uncertainties in the presented results

Despite these agreements, we note that there are some remaining uncertainties with respect to our trajectory calculations. For

example, due to the length of our back-trajectory calculations (up to 90 days), individual trajectories must not be analyzed,

nevertheless, statistical analyses are possible as noted by Bergman et al. (2013).

We further acknowledge that the TRJ calculations do not feature explicit convection. With respect to the importance of con-

vection, Wu et al. (2020) showed with a free-running CCM that for the first uplift in the ASM region convection is dominant

but in the UTLS the large scale dynamics are most relevant for the tracer budget. A recent study by Smith et al. (2021) inves-

tigated how convective processes are captured in the vertical velocity field of (re)analysis data. They came to the conclusion

that kinematic trajectories based on (re)analyses winds incorporate the effects of convection to a substantial degree. However,

they also noted that higher temporal and spatial resolution, e.g. as in ERA5 (Hersbach et al., 2020), seems to be favourable for

the inclusion of convective effects. Using the same modelling approach as in the present study, Bergman et al. (2013) showed

---

[1]Here we refer to the 1 degree data results of (Bergman et al., 2013) who find that about 35% of the PBL crossing trajectories, which in turn correspond to

roughly 78% of all trajectories starting in the AMA come from the TP in August 2011. This translates to an approximate contribution of the TP air masses to

the AMA of about 27%.

[2]As for the TP contribution the 1 degree values presented by Bergman et al. (2013) have been converted to contributions regarding all trajectories starting

within the AMA.



that their results from the kinematic trajectories regarding source region contributions are relatively robust with respect to the
choice of the resolution of the input data, which lends credit to their and also to our results. Further, they also found that the
vertical velocity of the (re)analysis data is correlated with observed precipitation data, which in turn is related to convective
activity.

   To assess the possible sensitivity of our results to missing convection, we also presented results from a free-running CCM
with Lagrangian transport and a convection parametrization, namely EMAC-ATTILA. These results suggest that seasonal
evolutions of some source regions (e.g. SEA and WP) are supported by the EMAC-ATTILA data and in particular the increase
of the TP contribution to the AMA air masses is also present in these data. This in turn indicates that for a qualitative description
of the contribution of these PBL source regions the explicit representation of convection might not be essential. Nevertheless,
in particular the fastest transport from the PBL to the upper troposphere might be underestimated in our TRJ data (cf. Figs. 5
and B2 - showing that trajectories are transported faster upward in the the EMAC-ATTILA data).

**5.3   Contribution of the TP**

Independently of potential limitations in the TRJ or EMAC-ATTILA data, the increase of TP air masses to the AMA compo-
sition is also backed up by ERA-Interim data, which is shown in Fig. 21: In May the core of the subtropical jet is located right
above the TP. During the course of the monsoon season, the tropical easterly jet, which is located on the southern boundary of
the AMA (Dethof et al., 1999), strengthens. This indicates an increase of the anticyclonic circulation of the AMA. Further, the
subtropical jet - which is located on the northern boundary of the AMA (Dethof et al., 1999) - as well as the zero-wind line
move northward. Consequently, air masses that are transported upward from the TP are likely to be advected by the subtropical
westerly jet during the early phase of the monsoon season (June), while they can feed into the core of the AMA during August.

   In this study, we investigated transport from the top of the PBL to the AMA, i.e. our analyses end at the top of the PBL.
Convergence of surface winds at the southern flank of the TP (Pan et al., 2016, their Fig. 8) might cause low level transport of
emissions from their source regions to the final exit and uplift region from the PBL to the AMA. As an example, emissions
e.g. of CO are low over the TP (Park et al., 2009; Barret et al., 2016, their Figs. 9 and 10, respectively), nevertheless air masses
transported from the PBL over the TP to the AMA can carry considerable CO signatures (Pan et al., 2016, their Figs. 2b and 7).
This issue is common to many of the previous studies regarding the source regions of AMA air masses and could be overcome
by employing surface emissions, which would lead to an analysis of the efficiency of transport to the AMA as noted already
for the use of forward trajectories by Bergman et al. (2013).

   Finally, as Bergman et al. (2013) found a relatively large contribution of air masses from the TP to the AMA, they discuss
their results in relation to other studies that either do or do not find important contributions of the TP to the air masses (or tracer
fields) in the AMA or UTLS. While they correctly argue that the results strongly depend on the chosen analysis method, we
can add another possible explanation for these differences: Most of the studies that find strong contributions of the TP to the
AMA or UTLS focus on August conditions e.g. Fu et al. (2006), Bergman et al. (2013) and Jensen et al. (2015).

   In contrast, Park et al. (2009) investigated the source region contribution and transport budget of CO to the AMA and came
to the conclusion that the TP has a relatively low impact on the CO maximum in the AMA region. For the source region





**Figure 21.** Zonal winds from ERA-Interim for May to August averaged over 1980 to 2009. Red (grey) colours indicate westward (eastward) winds and black contours indicate the zero-wind line. Grey shadings mark orography.

contribution, i.e. the contribution of CO emitted from the TP, they showed that the lack of surface emissions from the TP leads to this minor impact. In a vertically resolved CO budget analysis for the TP region they found that convection leads to a small
maximum around 400 hPa while advection leads to a negative tendency in the middle troposphere and thus argued that the TP does not play an important role with respect to CO transport to the AMA. The negative advection tendency found in their analysis is most likely related to the location of the subtropical jet over the TP in June 2005, which might have caused air





masses to be transported out of the TP region. In our analyses, the contribution from the TP to air masses within the AMA increases as the subtropical jet shifts northwards from June to August and we find, that the transport of TP boundary layer air out of the AMA region decreases accordingly (see Fig. 20).

Further, Devasthale and Fueglistaler (2010) put the importance of TP convection into perspective, however, they also showed that convective activity over the TP increases from June to August (see their Fig. 3). Similarly, from the convective upward mass flux in the EMAC-ATTILA data, we find that in July and August the mass flux into the upper troposphere (above ~350 hPa) over the TP is larger than in June (not shown). Hence, these previous results might also be strongly influenced by the different analysis periods.





# 6 Summary and conclusion

In this study we have analyzed the transport pathways and source regions from the PBL to the AMA. This was achieved by calculating trajectories for 14 monsoon seasons using reanalysis wind fields. Additional results from a Lagrangian transport model, which was run within a free-running CCM, were used to confirm these results. The presented analyses (Sects.3 and 4) and the discussion in the previous section (Sect.5) allow us to draw the following conclusions regarding the transport characteristics of air masses from the PBL to the AMA.

1. What is the climatological perspective of PBL to AMA transport in terms of pathways and PBL source regions? How reliable are previous results?

   – Our results show that during JJA on a climatological basis, AMA air mass sources come from a broad region in the PBL in Asia and with increasing height, the upward transport of air masses focuses on the eastern side of (or below) the AMA. However, by construction this analysis only captures the regions of upward transport to the AMA and not necessarily the full three-dimensional pathways. The main upward transport occurs on the eastern side below the anticyclone. We found an upward circling already considerably below 150 hPa for some of the trajectories. This result refines the original conduit schematic as depicted and discussed by Bergman et al. (2013) and is in full accordance with the large scale uplift above 360 K described by Vogel et al. (2019).

   The attribution of PBL source regions, however, is less clear. In TRJ, the largest contributions from the named source regions are found from the TP region (around 17%), the IND region (around 13%) and the WP region (around 12%). In LG-D we find almost the same contribution from the TP (15%) and the WP (12%), however the contribution from IND and SEA are the largest. This might be related to a large convective contribution, which is missing in TRJ. This could also imply that the convective contribution from WP is small, because both methods show the same contribution from WP. The LG-K results are similar to LG-D, but show a smaller contribution of the TP (9%).

2. How do the pathways and source regions vary on intraseasonal and interannual time scales?

   – With respect to the transport pathways, we find that the qualitative behaviour is similar throughout the monsoon season and between different monsoon seasons, i.e. upward transport on the eastern side below the AMA and subsequent upward transport within the AMA. Nevertheless, in particular with respect to the intraseasonal variation, the transport pathways shift considerably northwards over the course of the monsoon season in accordance with the shift of the monsoon system. With respect to the source regions, we find strong interannual and intraseasonal variability. For the latter, the contribution from the TP, which strongly increases from around 2% (4%) in TRJ (LG-D) in early June to around 24% (20%) in TRJ (LG-D) in early August, sticks out. This increase is in agreement with corresponding reanalyses data of the subtropical jet position. Considering the variability of the AMA is thus a potential starting point for reconciling differences in previous studies on PBL to AMA transport in particular with respect to the impact of the TP.





3. Are the PBL source regions and the transport pathways related to interannual east-west shifts of the AMA?

– We find shifts in the transport pathways between east and west years, although the main characteristics are qualitatively unchanged. Further, we find that the longitudinal shifts of the AMA are related to the so-called monsoon Hadley-Index. For the PBL sources we find no considerable differences between east and west years with respect to the defined source regions, while a map shows that there are (small) regional shifts in the contribution of the PBL sources.

From our results we find that the three-dimensional pathways of trajectories give a conclusive picture of transport from the PBL to the AMA. However, the relative contribution from the PBL source regions are (except for TP and WP) less robust. In our analysis we could not distinguish, whether the differences in source region contribution are a result of the different synoptic conditions in the free-running EMAC-ATTILA simulation compared to the reanalysis driven TRJ calculations or actually a result of the consideration of Lagrangian convection in the EMAC-ATTILA data. A first indication of faster vertical transport

due to parametrized convection in the LG data comes from the observation that a lower fraction of trajectories do not encounter the PBL in the LG simulations compared to the TRJ-data.

To allow for a more robust picture of the transport from the PBL to the AMA in the monsoon region, further investigations with various model setups would be beneficial. In particular, a set of tailored simulations with and without convective transport

would be valuable to assess the impact of convective transport with respect to individual source region contributions to AMA air masses.

*Data availability.* ERA-Interim data is available from the European Centre for Medium-Range Weather Forecast (ECMWF) (2011): (i) Copyright statement: Copyright "© [2022] European Centre for Medium-Range Weather Forecasts (ECMWF)". (ii) Source: www.ecmwf.int, (iii) Licence Statement: This data is published under a Creative Commons Attribution 4.0 International (CC BY 4.0). https://creativecommons.

org/licenses/by/4.0/. (iv) Disclaimer: ECMWF does not accept any liability whatsoever for any error or omission in the data, their availability, or for any loss or damage arising from their use. (v) Where applicable, an indication if the material has been modified and an indication of previous modifications: The trajectory data (TRJ) was derived using ERA-Interim data (ECMWF,2011). Further, additional analyses are based on ERA-Interim data.

**Appendix A**

**A1   EMAC-ATTILA setup details**

In the following, we provide a more detailed description of the EMAC-ATTILA simulation setup used in this study. This description is largely based on the description by (Brinkop and Jöckel, 2019) and we refer the reader to their publication for further details.





EMAC consists of the ECHAM5 (European Centre HAMburg general circulation model version 5; Roeckner et al., 2003, 2006)
model and the MESSy infrastructure (Modular Earth Submodel System Jöckel et al., 2005, 2010). Within EMAC, the submodel
ATTILA (see Reithmeier and Sausen, 2002, for the original model description) can be used to calculate Lagrangian transport
of air parcels (Brinkop and Jöckel, 2019). The simulations employed here are based on MESSy version 2.53 (corresponding
updates will be available in version 2.55) and are dubbed LG(diab) and LG(kin), respectively by (Brinkop and Jöckel, 2019).
We note that technically data from LG-D and LG-K were obtained from two different simulations, however the driving grid
point meteorology was identical in both cases (Brinkop and Jöckel, 2019).

Based on the description by Brinkop and Jöckel (2019) we present the most important aspects of these simulations here as
well: The simulations feature a spectral truncation for the ECHAM5 base model of T42 (corresponding to a quadratic Gaussian
grid of $\sim$2.8° x 2.8°) in the middle atmosphere (MA) setup with 47 model levels up to $\sim$0.01 hPa, i.e. the so-called T42L47MA
setup. In the free-running, i.e. no nudging of dynamic variables, EMAC-ATTILA simulations exploited here, radiatively active
trace gases have been prescribed from a previous simulation with full chemistry (RC1-base08 described by Jöckel et al., 2016)
to reduce the computational effort. Only a chemistry scheme for methane oxidation, the CH4 submodel (Winterstein and Jöckel,
2021), was turned on to allow for the feedback on stratospheric water vapour. Sea surface temperatures and sea ice cover in
these simulations were prescribed from HadISST (Rayner et al., 2003).

**A2  AMA boundary determination**

In this study mostly trajectories starting within the core of the AMA have been analyzed. The determination of the boundary
of the AMA is difficult and many studies have used various quantities and thresholds to determine the boundary of the AMA
(e.g. Park et al., 2007; Garny and Randel, 2013; Ploeger et al., 2015; Santee et al., 2017). Here, the boundary determination is
based on a geopotential height anomaly (GPHA) threshold as proposed by Barret et al. (2016). They calculated GPHAs with
respect to the 50° S-50° N mean and used a threshold of 270 m for the pressure levels 100, 150 and 200 hPa based on previously
used boundaries. For our data, we have derived thresholds explicitly for the trajectory model calculations using ERA-Interim
data at 2.5 degree grid spacing and for the EMAC-ATTILA simulations using the CCM grid point data. In principal we have
determined suitable threshold candidates by producing scatter plots of the geopotential height anomaly and the maximum
meridional wind strength along the ridge-line of the AMA (see Zhang et al., 2002, for the ridge-line definition). For EMAC-
ATTILA, we further required the maximum wind speed to be located at a grid point with GPHA of at least 100 m to avoid noise
from unrealistically low values. Using this technique, we determined anomaly thresholds of 280 m and 295 m for ERA-Interim
and EMAC-ATTILA data, respectively. The value of 280 m for ERA-Interim is in good agreement with the threshold of 270 m
used by Barret et al. (2016).

**A3  Selection of summer seasons for the TRJ calculations**

The trajectory model calculations described in Sect. 2 have been performed for 14 NH-summer seasons in the period 1979–
2013. These NH-summers have been selected as the mean position of the AMA was rather displaced to the East or West. In
detail, a modified version of the South Asian High Index (SAHI), which was originally defined by Wei et al. (2014), has been





used. Wei et al. (2014) calculated the SAHI by standardizing the time series of differences of geopotential height over a box in the east of the AMA (22.5-32.5°N x 85-105°E) minus that over a box in the west of the AMA (22.5-32.5°N x 55-75°E) at a single pressure level. Compared to the definition by Wei et al. (2014), we use a modified version, which standardizes the sums

of these differences over three pressure levels (100, 150 and 200 hPa). We use these pressure levels as they are centered around the starting level of the trajectories (150 hPa). ERA-Interim data with a grid spacing of 2.5° x 2.5° have been used to determine the modified SAHI and using a threshold of ±0.7 deviation from the mean we found fourteen years with a rather eastward or westward displaced AMA (seven years each).[3] The corresponding starting probabilities for the east (cyan) and west (purple) composites are shown in Fig. 2.

**Appendix B: Supplemental figures**

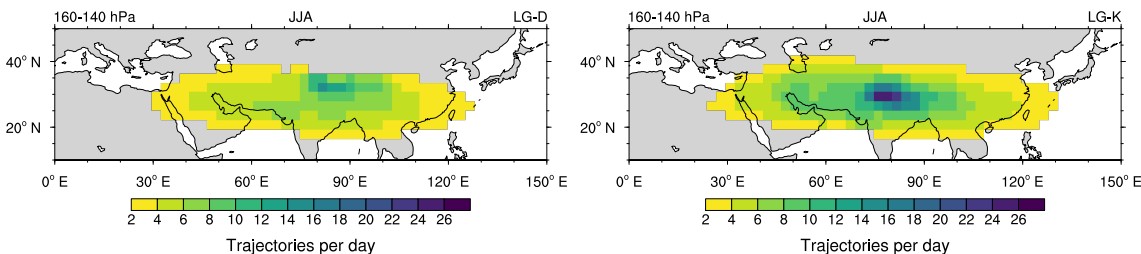

**Figure B1.** Starting frequency of trajectories for LG-D (left) and LG-K (right) over the years 1981-2010. For LG-K data for 2008 was removed - see text for details.

---

[3]West years: 1980, 1984, 1994, 2001, 2007, 2008 and 2011 – East years: 1981, 1987, 1989, 1998, 2009, 2010 and 2012.

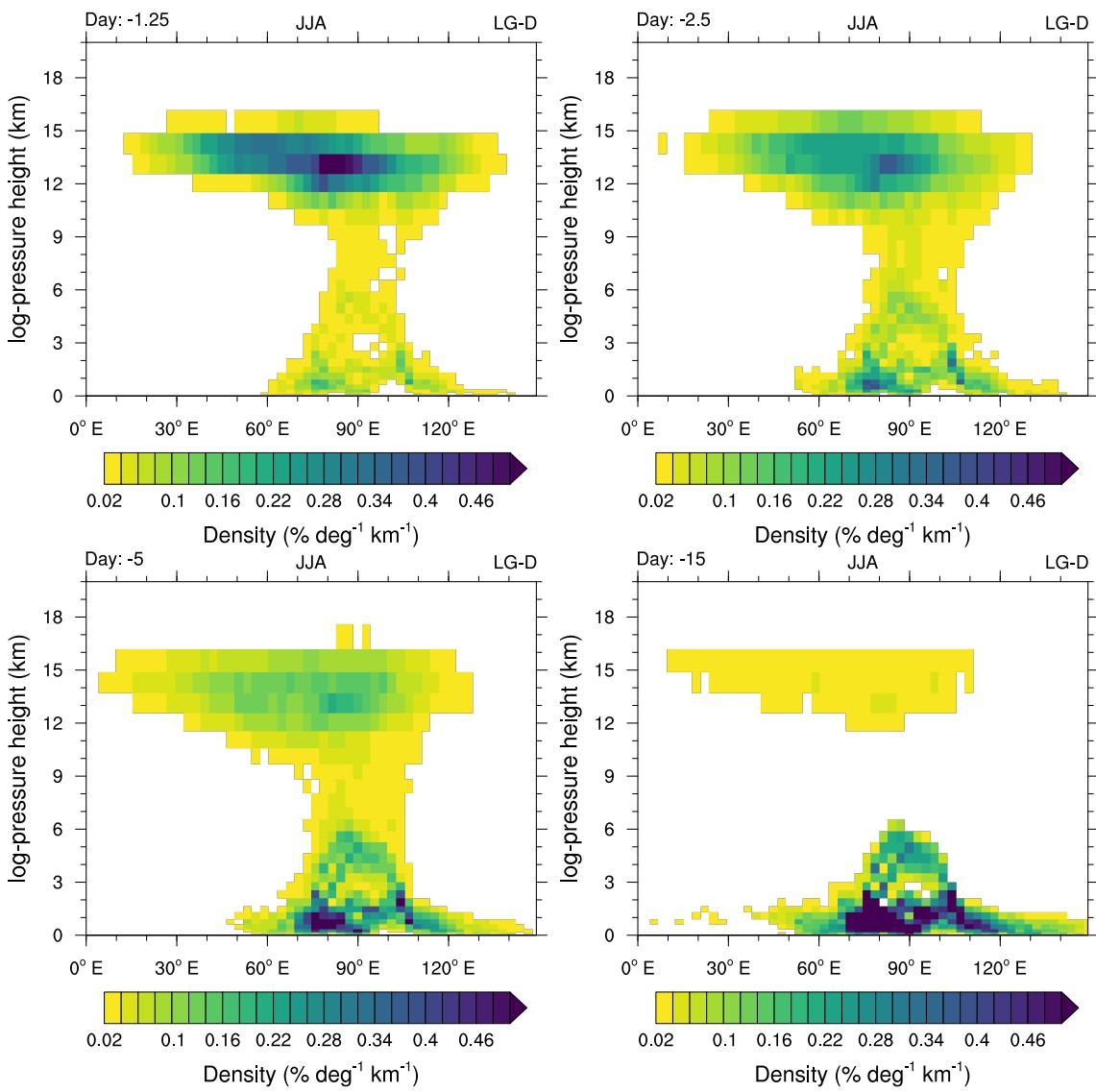

**Figure B2.** Density of trajectory distributions integrated over 0-50° N as in Fig 5 but for the LG-D data from 1981-2010 for 1.25, 2.5, 5 and 15 days prior to the arrival of the trajectories in the AMA at approximately 150 hPa.





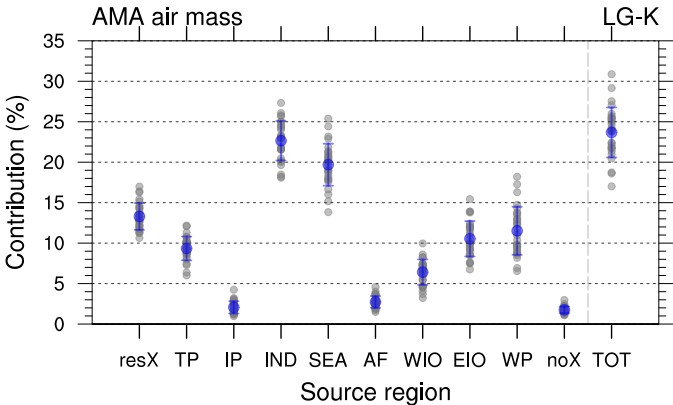

**Figure B3.** Contributions of PBL sources to the AMA around 150 hPa for the LG-K simulation for 1981-2010 (with 2008 removed, see text for details). Whiskers denote the interannual standard deviation, whereas individual years are indicated via grey dots. For TOT 1% corresponds to 8000 trajectories.

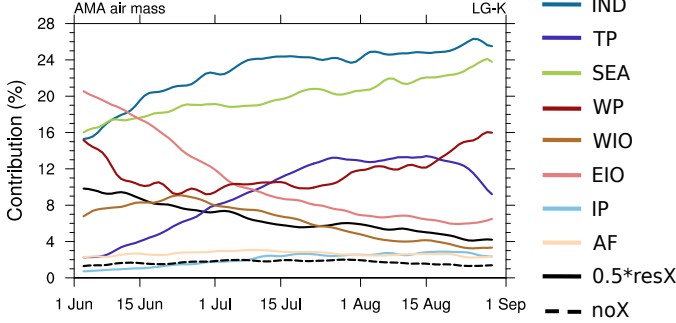

**Figure B4.** Source evolution in the LG-K data during 1981 to 2010 (with 2008 removed, see text for details). resX data has been scaled by 0.5. All contributions have been smoothed via 5 day running means (weights of $[\frac{1}{9}, \frac{2}{9}, \frac{3}{9}, \frac{2}{9}, \frac{1}{9}]$), while daily data were produced from summing up the ten hourly data for each day.





*Author contributions.* Large parts of the work presented here - in particular the kinematitc trajectory analyses - are based on work performed for the PhD thesis of MN. The presented analyses have been performed by MN with help of SB for postprocessing of the EMAC-ATTILA data. The EMAC-ATTILA simulations were performed by SB and PJ. The manuscript was mainly composed by MN, while all authors contributed to the writing and discussion.

*Competing interests.* In accordance with the competing interest policy of Copernicus Publications, we note that two (or more) co-authors are members of the editorial board of ACP. The authors declare that no other conflict of interest is present.

*Acknowledgements.* We thank Helmut Ziereis (DLR) for comments on the manuscript. We thank the ECMWF for producing and distributing ERA-Interim reanalysis data. EMAC-ATTILA data were produced and analyzed at the Leibniz-Rechenzentrum in Garching, Germany. The authors gratefully acknowledge the Gauss Centre for Supercomputing e.V. (www.gauss-centre.eu) for funding this project by providing
computing time on the GCS Supercomputer SuperMUC-NG at Leibniz Supercomputing Centre (www.lrz.de). CDO (Climate Data Operators) was used for data processing (available at https://code.mpimet.mpg.de/projects/cdo/, last access: 24 Jan. 2022; Schulzweida, 2021)). We used the NCAR Command Language (NCL, 2019, see references) for data analysis, processing and graphics.

*Financial support.* The research leading to these results has received funding from the European Community's Seventh Framework Programme (FP7/2007–2013) under grant agreement no. 603557. The work described in this paper has received funding from the Initiative and
Networking Fund of the Helmholtz Association through the project "Advanced Earth System Modelling Capacity (ESM)". The content of the paper is the sole responsibility of the author(s) and it does not represent the opinion of the Helmholtz Association, and the Helmholtz Association is not responsible for any use that might be made of the information contained. HG was funded by the Helmholtz Association under grant no. VH-NG-1014 (Helmholtz Young Investigators Group MACClim). M. Park acknowledges support from the NASA's Aura Science Team Program (NNH19ZDA001N-0030). L. Pan acknowledges support from the US National Science Foundation (NSF) through
the funding of National Center for Atmospheric Research (NCAR) operation, grant AGS-1852977.





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
