# Peer review of "Climatology and variability of air mass transport from the boundary layer to the Asian monsoon anticyclone"

_Atmospheric Chemistry and Physics, 2022_

## Referee Comment (RC1)

Review of *Atmospheric Chemistry and Physics* manuscript 10.5194/acp-2022-143 by Nützel et al.: *Variability of air mass transport from the boundary layer to the Asian monsoon anticyclone*

**General comments**

1. Line 109: For trajectory calculations involving deep convection, both the space and time resolution of the wind fields are important. The 6-hour time resolution, in particular, and the 1.5° horizontal resolution of the ERA-Interim data are both rather problematic for calculating 'convective' transport. Equally significant is the hydrostatic nature of the underlying atmospheric model. While the total vertical mass flux due to convection may be roughly correct, the fact that the reanalysis system is based on a global hydrostatic model means that the vertical velocities are too small, probably by an order of magnitude or more, and occur over too large an area.

   The ERA5 reanalysis, which has been available for several years, has higher spatial and, more importantly, temporal resolution. (The authors note related issues in §5.2.). I recommend doing a test calculation (e.g., one season) to compare ERA5 trajectories with the ERA-Interim trajectories. If the results are similar, it would not be necessary to re-run all of the trajectories and the analysis. If not, the calculations should be re-done using the newer ERA5 reanalysis.

2. §2.2: Were the EMAC trajectory calculations done 'online', that is, with a time step equal to the model time step? What is the model time step? Why were the EMAC data output at 10 h intervals? That is an odd choice and could cause some unusual aliasing of the diurnal cycle.

3. Figures 3 and 12: I do not understand why the crossing maps at lower altitudes (e.g., 400 hPa and $\eta = 0.85$) bear so little resemblance to the distribution of monsoon precipitation, which is directly related to vertical motion and diabatic heating. The heaviest precipitation, which is strongly correlated with the occurrence of deep convection, is located along the west coast of India, the east coast of the Bay of Bengal, the northern Philippine Islands, and the Himalayan front. None of these features, except possibly the Bay of Bengal, show up in the transport from the PBL. The patterns of upward transport also differ from the GPM radar echo-top climatology (Liu and Liu, JGR, 2016). Have you compared the precipitation distributions in ERA-Interim and the EMAC model simulations with observations (e.g., TRMM TMPA)?

   At higher levels the ascent is presumably due to radiative rather than latent heating, so the difference from the precipitation distribution is easier to explain.

4. §3.1.2: By 'boundary layer source regions' do you mean the regions where the trajectories ascend out of the PBL (in the forward direction)? Air can spend a long time in the boundary layer and move from one region to another within the boundary layer before being entrained in a convective updraft and lofted out of the boundary layer.

5. §4 and Figure 18: The model results show much larger contributions from the IND and SEA regions and less from the TP, which corresponds better to the observed precipitation distribution.

6. The text is rather verbose and repetitive, and as a result the paper is longer than it needs to be. This can be corrected by thorough editing.

**Minor comments**

1. Title: The paper does address variability of transport to some extent, but the main focus is on the mean transport.

2. Line 54: How is ascent 'driven by the large-scale anticyclonic circulation'? Ascent in an isentropic sense must be driven by diabatic heating, which at these altitudes must be due primarily to net radiative heating.

3. Line 86: The sentence beginning 'Results from this model ...' is not clearly written.

4. Line 91: This paragraph is unnecessary and can be deleted.

5. Figure 5: Please add a pressure scale to the plots.

6. Figures 10 and 11: Can you combine these two figures into one (for easier comparison) or simply eliminate Figure 10? There is little difference between them.

7. Figure 15: Since you are plotting the relative contributions from different regions, the figure might be easier to follow if you plot the cumulative amounts across the regions (i.e., a stacked plot).

8. Figure 16: This figure does not add much information to what has already been presented in Figures 10, 11, and 15. I suggest removing it, or at least combining it with Figures 10 and 11.

9. Figure 17: It is difficult to flip back and forth between Figures 10 and 17 in order to compare them. These plots really belong in the same figure.

10. Figures 18 and 19: As with the box and whisker plots, it is difficult to compare these results with Figure 15. These should all be in one figure.

11. §6: This section is longer than necessary. A short statement of the principal results would be sufficient.

12. Appendix A: This appendix adds little information to what is already presented in §2.2.

**Recommendation**

This paper presents an analysis of vertical transport to the upper troposphere and lower stratosphere within the Asian summer monsoon circulation. The manuscript is rather long considering that the results largely confirm earlier studies (e.g., Garny and Randel; Bergman; and Vogel) while adding some new details.

The two main issues that I see with the manuscript are:

1. The ERA-Interim reanalysis has been succeeded by the ERA5 reanalysis. ERA5 offers improved spatial and temporal results, which could affect the trajectory calculations enough to change the results. The authors should compare trajectories from ERA-Interim and ERA5 to ensure that their results would not be affected significantly by switching to ERA5.

2. Scientifically my main concern with the manuscript is that the patterns for ascent of the air parcels do not correspond well to the observed locations of heavy precipitation and deep convection across the Asian monsoon region. The trajectories could be correct (in the sense that they are representative of the real world), and there could be a physical explanation for why the regions of ascent are displaced from the convection, but it could also indicate a systematic problem with the reanalysis, such as vertical ascent much slower than actual updraft speeds so that ascent occurs far from the convection. The latter would not be surprising given the hydrostatic nature of the reanalysis system model and the necessity for highly idealized convective parameterizations.

I recommend publication after addressing these two points.

---

## Author Comment (AC1)

**Author Comment to manuscript ACP-2022-143 (https://doi.org/10.5194/acp-2022-143, in review, 2022): "Variability of air mass transport from the boundary layer to the Asian monsoon anticyclone"**

by M. Nützel et al.

June 23, 2022

We thank the referees for taking time to review our paper and appreciate the referees' efforts to improve the manuscript. In the following we address each review comment (*black italics*) by stating our reply (blue). In addition we appended a manuscript version which highlights the changes between the ACPD version and the revised version.

**Reply to comments from Referee #1 (https://doi.org/10.5194/acp-2022-143-RC1)**

Below we will address all comments of referee #1 and will state corresponding changes in the manuscript. Again, we would like to thank referee #1 for taking the time to review our manuscript.

*Review of Atmospheric Chemistry and Physics manuscript 10.5194/acp-2022-143 by Nützel et al.: Variability of air mass transport from the boundary layer to the Asian monsoon anticyclone*

**General comments**

*1. Line 109: For trajectory calculations involving deep convection, both the space and time resolution of the wind fields are important. The 6-hour time*

*resolution, in particular, and the 1.5° horizontal resolution of the ERA-Interim*
*data are both rather problematic for calculating 'convective' transport. Equally*
*significant is the hydrostatic nature of the underlying atmospheric model. While*
*the total vertical mass flux due to convection may be roughly correct, the fact*
*that the reanalysis system is based on a global hydrostatic model means that the*
*vertical velocities are too small, probably by an order of magnitude or more, and*
*occur over too large an area. The ERA5 reanalysis, which has been available*
*for several years, has higher spatial and, more importantly, temporal resolution.*
*(The authors note related issues in §5.2.). I recommend doing a test calculation*
*(e.g., one season) to compare ERA5 trajectories with the ERA-Interim trajec-*
*tories. If the results are similar, it would not be necessary to re-run all of the*
*trajectories and the analysis. If not, the calculations should be re-done using the*
*newer ERA5 reanalysis.*

Reply: We agree with the reviewer that for many aspects higher temporal and
spatial resolution is favourable. We address this issue in the discussion (Sect. 5.2)
by referring to the study by Smith et al. (2021). However, we also note that
this is a rather general issue that applies to many problems in our field. Here,
we would like to point out that the storage of input and output data as well as
the calculation of the trajectories is an issue that needs to be taken into account
when conducting such experiments. Our explicit focus was on trajectory studies
for many years - and not sensitivities with respect to the reanalysis product or
the temporal/spatial resolution. Acquiring the input data for ERA5 (higher
temporal and spatial resolution) alone would have been a huge effort. As to the
one year sensitivity: using any other reanalysis data (or resolution) would likely
influence the quantitative results, however, we assume that the qualitative re-
sults would still hold. Such a sensitivity is beyond the scope of our study and as
mentioned in the text has been conducted by Bergman et al. (2013). They come
to the conclusion that concerning the PBL contributions, when only accounting
for PBL crossing trajectories, the effect is relatively limited. We want to point
out that we show the results from the free-running EMAC-ATTILA simulation
which features the impact of (simulated) convection explicitly. Further, we em-
phasize that the results from Legras and Bucci (2020) for 2017 with respect to
their so-called convective impacts from ERA-Interim and ERA5 data show sim-
ilar features as our boundary layer source maps (see definition of boundary layer
source as reply to your general comment #4). To our understanding the issue of
the hydrostatic model would remain for ERA5 as in Section 4 in Hersbach et al.
(2020) no transition to non-hydrostatic modelling is mentioned. We also note

that comparability with previous studies is an issue and as ERA-Interim has been used often and we had to use ERA-Interim in a related project (because of the mentioned data storage issues), there are also advantages of using ERA-Interim. We further want to note that the reviewer's scepticism with respect to the ERA-Interim trajectory results is likely also related to the reviewer's general remark #3, which we clarify below.

*2. §2.2: Were the EMAC trajectory calculations done 'online', that is, with a time step equal to the model time step? What is the model time step? Why were the EMAC data output at 10 h intervals? That is an odd choice and could cause some unusual aliasing of the diurnal cycle.*

Reply: Yes, the EMAC trajectory calculations were done online with a model time step of 600 s using the submodel ATTILA (Brinkop and Jöckel, 2019). In the revised version a sentence was slightly modified to be more precise: "Within these two EMAC-ATTILA simulations - which have the same grid point meteorology - about 1.16 million air parcels, which represent the global atmosphere, are initialized once at the beginning of the simulation and are consequently transported online with a model time step of 600 s according to the CCM's meteorological fields (Brinkop and Jöckel, 2019)." The "odd" output interval is actually chosen on purpose: The EMAC-ATTILA simulations were not specifically designed for this study and it is common in our simulations to write output data every 10 hours. This is done to capture every second hour of the day (every once in a while). This choice is made to have a reasonable representation of the diurnal cycle and to get better temporal averages in a long-term statistical sense, while limiting the output.

*3. Figures 3 and 12: I do not understand why the crossing maps at lower altitudes (e.g., 400 hPa and $\eta = 0.85$) bear so little resemblance to the distribution of monsoon precipitation, which is directly related to vertical motion and diabatic heating. The heaviest precipitation, which is strongly correlated with the occurrence of deep convection, is located along the west coast of India, the east coast of the Bay of Bengal, the northern Philippine Islands, and the Himalayan front. None of these features, except possibly the Bay of Bengal, show up in the transport from the PBL. The patterns of upward transport also differ from the GPM radar echo-top climatology (Liu and Liu, JGR, 2016). Have you compared the precipitation distributions in ERA-Interim and the EMAC model simulations with observations (e.g., TRMM TMPA)? At higher levels the ascent*

99 *is presumably due to radiative rather than latent heating, so the difference from*
100 *the precipitation distribution is easier to explain.*

101 Reply: We agree that at first this difference can seem disturbing. However, we
102 want to point out that our analysis is conditioned on trajectories that reach the
103 AMA at 150 hPa. This means we only analyse air masses that find their way to
104 the AMA at 150 hPa. Maps showing precipitation patterns do not have these
105 restrictions. The discrepancy between precipitation maps and source maps has
106 already been noted by Legras and Bucci (2020) (see end of their section 3.1) and
107 also Bergman et al. (2013) touch on this subject (see their Fig. 7 and section 5).
108 We note that precipitation maps from observations (e.g. Xie et al., 2006, their
109 Fig. 1) also do not directly correspond to high cloud distributions in the Asian
110 monsoon region as shown by Devasthale and Fueglistaler (2010). Further, it is
111 noted by Shige and Kummerow (2016) that orographic precipitation over west
112 India is often related to low clouds. Based on these previous studies and our
113 analyses, our understanding is as follows: low- to mid-level convection might
114 be important for the precipitation patterns but air parcels that are transported
115 upwards in this convection need to find a region of onward transport to the
116 AMA. Seemingly, for some of the regions with heavy precipitation this rarely
117 happens. Finally, the maps of convective impact shown by Legras and Bucci
118 (2020) show similar patterns as our analyses, despite the different modelling
119 approaches. This lends further credit to the consistency of our analyses.

121 *4. §3.1.2: By 'boundary layer source regions' do you mean the regions where*
122 *the trajectories ascend out of the PBL (in the forward direction)? Air can spend*
123 *a long time in the boundary layer and move from one region to another within*
124 *the boundary layer before being entrained in a convective updraft and lofted out*
125 *of the boundary layer.*

126 Reply: Yes, we account for the last crossing points of trajectories with the top
127 of the PBL, i.e. starting from the initialisation and going back in time, we note
128 where the trajectory first encounters the top of the PBL. We point that out
129 more clearly in the revised version to avoid any confusion. For example, in sec-
130 tion 2.3 we now write: "When the pressure at the trajectory position is larger
131 than 0.85 times the surface pressure below the trajectory, we assume that the
132 trajectory has encountered the PBL as described by Bergman et al. (2013). The
133 first location where this happens backward in time will be referred to as bound-
134 ary layer source of the trajectory." Additionally, at some instances we changed
135 "from the PBL" to "from the top of the PBL" and we changed the wording in

the last paragraph of the introduction of the revised version to: "...are followed backward in time to their first crossing of the top of the PBL...". Further, we agree with the referee and we note that we addressed this issue in the discussion (L483-490 in the ACPD version).

*5. §4 and Figure 18: The model results show much larger contributions from the IND and SEA regions and less from the TP, which corresponds better to the observed precipitation distribution.*

Reply: As outlined in our reply concerning your general comments #2, the precipitation distribution does not have to match with the boundary layer source distributions. In accordance, Legras and Bucci (2020) show strong convective impacts from the Tibetan Plateau at and above approx. 360 K with their combined reanalysis/observation modelling approach. Moreover, we have veryfied that the 2D PBL source distribution looks similar for EMAC-ATTILA (not shown) as for the TRJ data, with the main difference that the contribution of the Tibetan Plateau is less pronounced. The differences between EMAC-ATTILA and the TRJ data data are discussed in the lines 360-364 in the ACPD version.

*6. The text is rather verbose and repetitive, and as a result the paper is longer than it needs to be. This can be corrected by thorough editing.*

Reply: We shortened the paper and made it more concise. For example, the text in Section 2 before Section 2.1 was partly (re-)moved, the Appendix A1 was deleted and parts from Section 3 have been deleted or shifted to Section 5 and vice-versa.

**Minor comments**

*1. Title: The paper does address variability of transport to some extent, but the main focus is on the mean transport.*

Reply: We think that we present a number of analyses showing interannual and intraseasonal variability, e.g. Figs. 2, 7-8, 10-19 of the ACPD version contain information regarding interannual or intraseasonal variability. Of course, we also present many climatological views, which we see as a prerequisite to be able to address interannual and intraseasonal variability. To account for the fact that we present this climatological perspective (as stated in the abstract of the ACPD version), we changed the title to: "Climatology and variability of air mass transport from the boundary layer to the Asian monsoon anticyclone".

2. *Line 54: How is ascent 'driven by the large-scale anticyclonic circula-tion'? Ascent in an isentropic sense must be driven by diabatic heating, which at these altitudes must be due primarily to net radiative heating.*

Reply: We thank the reviewer for spotting this error: "driven" should rather be "follows". We changed the text accordingly.

3. *Line 86: The sentence beginning 'Results from this model ...' is not clearly written.*

Reply: Is changed to "Results from the Lagrangian model ..."

4. *Line 91: This paragraph is unnecessary and can be deleted.*

Reply: As per the reviewer's request, the paragraph containing the manuscript's outline was deleted. The references to Sections 3 and 4 have been shifted to the paragraph above.

5. *Figure 5: Please add a pressure scale to the plots.*

Reply: We have thought about adding a pressure scale to the plots Figs. 5, 6, 8, 14 and B2 (ACPD version). However, we decided against it, for the fol-lowing reasons: a) the densities of the trajectory positions have exactly been constructed with log-p height as vertical axis and hence the corresponding units contain the factor $km^{-1}$, b) the busy figures would get more busy with no real information added as, c) the conversion from log-p height to pressure is straight forward (see updated Figure caption).

6. *Figures 10 and 11: Can you combine these two figures into one (for easier comparison) or simply eliminate Figure 10? There is little difference between them.*

Reply: We have combined Figs. 10 and 11 in the revised manuscript.

7. *Figure 15: Since you are plotting the relative contributions from different regions, the figure might be easier to follow if you plot the cumulative amounts across the regions (i.e., a stacked plot).*

Reply: We have thought about such a plot, however, we think it is sometimes harder to actually tell the exact quantities as the base for each source region would then vary. Hence we opted for single lines relative to zero.

8. *Figure 16: This figure does not add much information to what has al-*

*ready been presented in Figures 10, 11, and 15. I suggest removing it, or at least*
*combining it with Figures 10 and 11.*

Reply: We decided to keep the figure as no interannual variability is given in Fig. 15, whereas it is presented in 16. Figs. 10 and 11 do not show the individual variability of the PBL source contributions according to the different months (June, July and August). The respective text has been shortened and the figure is now combined with the previous Fig. 15.

*9. Figure 17: It is difficult to flip back and forth between Figures 10 and 17*
*in order to compare them. These plots really belong in the same figure.*

Reply: As we have already combined Figs. 10 and 11 as the reviewer suggested, we do not see the option to add another data set here. The plots will get too crowded. Further, we agree that the comparison would be easier if everything is in the same figure as subplots. However, we think it is more important to distinguish between the data sets as our focus lies on the TRJ data. Keeping the analyses for EMAC-ATTILA data separate from the TRJ data avoids mixing up the results and is in accordance with the structure of the text, i.e. first the results from the TRJ data and then the results from EMAC-ATTILA. Nevertheless, we included the TRJ results as faint blue dots and whsikers to facilitate the comparison.

*10. Figures 18 and 19: As with the box and whisker plots, it is difficult to*
*compare these results with Figure 15. These should all be in one figure.*

Reply: We combined Figs. 18 and 19, however, we kept them separate and also separate from the TRJ results. See also our reply to your minor comment 9.

*11. §6: This section is longer than necessary. A short statement of the*
*principal results would be sufficient.*

Reply: We shortened the respective section, however, we would like to keep the structure of answering our question from the introduction.

*12. Appendix A: This appendix adds little information to what is already*
*presented in §2.2.*

Reply: We assume that you are referring to the section A1 as this section corresponds to section 2.2. Hence, we rephrased Section 2.2 and removed the Appendix A1.

**Recommendation**

*This paper presents an analysis of vertical transport to the upper troposphere and lower stratosphere within the Asian summer monsoon circulation. The manuscript is rather long considering that the results largely confirm earlier studies (e.g., Garny and Randel; Bergman; and Vogel) while adding some new details. The two main issues that I see with the manuscript are:*

*1. The ERA-Interim reanalysis has been succeeded by the ERA5 reanalysis. ERA5 offers improved spatial and temporal results, which could affect the trajectory calculations enough to change the results. The authors should compare trajectories from ERA-Interim and ERA5 to ensure that their results would not be affected significantly by switching to ERA5.*

Reply: Please consider our reply concerning your general comment #1. We assume, that the scepticism regarding our results is likely also related to the second recommendation of the reviewer. Taking our reply with respect to that comment into account, we do not see any indications of inconsistencies. Of course the quantitative results will change using a different reanalysis or resolution, but the main qualitative results will likely be robust.

*2. Scientifically my main concern with the manuscript is that the patterns for ascent of the air parcels do not correspond well to the observed locations of heavy precipitation and deep convection across the Asian monsoon region. The trajectories could be correct (in the sense that they are representative of the real world), and there could be a physical explanation for why the regions of ascent are displaced from the convection, but it could also indicate a systematic problem with the reanalysis, such as vertical ascent much slower than actual updraft speeds so that ascent occurs far from the convection. The latter would not be surprising given the hydrostatic nature of the reanalysis system model and the necessity for highly idealized convective parameterizations.*

Reply: Please consider our comments regarding your general comment #3. In particular, that high clouds, which partly might effectively feed into the AMA and precipitation maps do not necessarily have to align. Again, we want to stress that Legras and Bucci (2020) find similar distributions for their analysis of convective impact at and above approx. $360\,\text{K}$ based on ERA5 reanalysis and observational cloud data. Hence, although the distributions of precipitation and source regions are different, there is no scientific inconsistency.

284    *I recommend publication after addressing these two points.*

285    Reply: We hope, that we have been able to sufficiently address the reviewer's

286    comments.

287

**Reply to comments from Referee #2 (https://doi.org/10.5194/acp-2022-143-RC2)**

Below we will address all comments of referee #2 and will state corresponding changes in the manuscript. Again, we would like to thank referee #2 for taking the time to review our manuscript.

*The paper analyses the PBL sources and the pathways of transport in the AMA UTLS region at climatological level, by use of multiannual back-trajectories and, to understand the convection contribution, CCM simulations.*

*General comments:*

*The paper gives an exhaustive view of the transport processes in the region, it's well written, structured and the figures are well presented. The major problem of this paper lies in its verbosity and repetitiveness, which makes the manuscript extremely long and dispersive. I would therefore encourage the paper for publication, after some editing and after addressing some minor points.*

Reply: We thank the reviewer for the positive feedback regarding the general presentation of the mansucript. We made the presentation more concise in our revised version. Some of the requested changes from reviewer #1 aim at the same direction. Below, we will reply to all comments made by the reviewer.

*Specific comments: The abstract is one particular example of a section that needs to be more concise. It should rather focus on the main points that the authors think the paper is addressing without diluting with too many unnecessary details!*

Reply: We shortened the abstract by slightly rephrasing it.

*Similarly, between the Introduction and the Data and methods sections, there are several repetitions on the models description and how they will be used.*

Reply: We have shortened the Introduction as suggested by reviewer #1. Further, we restructured Section 2 with the aim to reduce repetitions and be more concise.

*Line 118: The authors say "Therefore" a modified version of the so-called SAHI index has been used. It would be useful to have a short explanation of what the SAHI is and a more precise explanation of which are the reasons why*

*323*  *it has to be modified for the purposes of this analysis.*

324 Reply: The corresponding section was rephrased and moved to 2.3.1. It now
325 reads: "For the selection a modified version of the so-called South Asian High
326 Index (SAHI; Wei et al., 2014), which measures the east–west displacement of
327 the AMA, has been employed. The modification, which uses the geopotential
328 height at three pressure levels - compared to one as originally defined by Wei
329 et al. (2014) - is supposed to better capture the 3D structure of the AMA. A
330 detailed explanation for the choice of the years and a description of the selection
331 process is given in the Appendix A2." We hope that the description is clearer
332 and easier to follow now.

333

*334*  *Line 152: What does it mean by "Pressure below the trajectories"? Is it the*
*335*  *pressure right below the lowest trajectories or right below the mean position of*
*336*  *the trajectories? Or the mean value of the pressure in the whole layer below the*
*337*  *trajectories?*

338 Reply: Thank you for the comment. The statement was unclear. It is cor-
339 rected in the revised version: "When the pressure at the trajectory position is
340 larger than 0.85 times the surface pressure below the trajectory, we assume that
341 the trajectory has encountered the PBL as described by Bergman et al. (2013)."

342

*343*  *Line 160: It is not clear to me how the choice of the 295m threshold value*
*344*  *for the AMA has been made. Is it by comparing the AMA boundaries shape with*
*345*  *what obtained from ERA-Interim data?*

346 Reply: To avoid a lengthy description in the text, we referred the reader to the
347 Appendix A2. As the previous description was misleading, it has been updated
348 in the revised version and we hope that the description is easier to follow now.
349 The corresponding part in the Appendix (A1 of the revised version) now reads:
350 "...In principal, we have determined suitable threshold candidates by deriving a
351 single GPHA value, which on average represents the strongest anticyclonic cir-
352 culation. This was done by calculating the mean of the GPHA values associated
353 with the strongest meridional winds (southward and northward) along the ridge
354 line (see Zhang et al., 2002, for the ridge line). For EMAC-ATTILA, we further
355 required the maximum wind speed to be located at a grid point with GPHA of
356 at least 100 m to avoid noise from unrealistically low values. Using this tech-
357 nique, we determined approximate anomaly thresholds of 280 m and 295 m for
358 ERA-Interim and EMAC-ATTILA data, respectively. The value of 280 m for
359 ERA-Interim is in good agreement with the threshold of 270 m used by Barret et al. (2016)." Additionally, for EMAC-ATTILA we have also checked, that the climatological AMA associated with the threshold of 295 m looks reasonable.

*Line 176: The authors compare the 14 years trajectories analysis with the 1981 to 2010 one from the CCM. As the 14 trajectories years has been chosen among the more westward and more eastward shift years of the AMA, I was wondering if it is really representative of the climatology of the period. In addition, are the differences between the CCM and the trajectories analysis related mostly to the convective activity or may be related to the transport behaviour of air masses during the non-considered years?*

Reply: A year to year comparison is not possible as the CCM is free-running (see respective text). With respect to the choice of the 14 years: as the East/West years show some differences but the main paths are similar and the discrepancies between the source region contributions are rather small, we assume that the full climatology would not look different. Further, we also point out that the main points of the paper are robust. The difference between CCM and TRJ are likely attributable to two factors: a changed background dynamic and the effect of parametrized convection. A clear separation is not possible from our data and additional simulations and analyses would be needed to distinguish the convective impact (see Summary and Conclusion).

*Line 213: I would suggest choosing a different wording than "re-circulation", which recall more the horizontal recirculating patter in the AMA rather than the vertical displacement.*

Reply: Actually, what is meant here is a mixture between both: horizontal circulation within the AMA and vertical upward (downward) movement on the eastern (western) side. The later results in a net upward movement and the full pathway is described as "upward spiraling" by Vogel et al. (2019). Anyhow, the respective sentence has been changed in the revised version.

*Caption figure 8: can you rephrase the "will be noted at the crossing point also later in time"? It's not clear what you mean with that.*

Reply: If a trajectory reaches the PBL it is noted in the analyses at that crossing position, i.e. the position where it first encountered the PBL, also for time points further back in time. As this procedure already applies to the analysis presented in Fig. 5 (ACPD, Fig. 6 in the revision), we rephrased the wording in the corresponding figure caption: "Once trajectories reach the PBL their

pathways are not followed back any further. Instead, they are noted at their first PBL-crossing points also for analyses going back further in time. For example, if a trajectory reaches the PBL already after 3 days, it will be counted at this PBL-crossing position also for the analysis 5 days and 15 days back in time." In the figure caption of Figs. 6/8 (ACPD, Fig. 7/9 in the revised version), we write now: "Once trajectories reach the PBL they are not tracked further and will be noted at the crossing point also further back in time (as in Fig. 6)."

*Line 255: Why here you choose 2 km and in the figure 3 km as a threshold for the TP?*

Reply: We thank the reviewer for spotting this issue. The analysis have all been performed with respect to the 2 km threshold. The outlines of the TP via the 3 km threshold in Figs. 1 and 2 (ACPD version) were given for orientational purposes only. However, to avoid any confusion, in all figures the TP is shown via 2 km contour now. Further, the contours are now also described in Fig. 1 (Fig. 2 revised version; see also our reply to the comment concerning "Caption Figure 1").

*Figure 10 and similar: I had some problems understanding how to read the TOT variable. Is it really a percentage (the % of the total trajectories who start in the AMA) or it is just a way to represent the total number of trajectories by the 1 to 4000 conversion? As it's in the same plot as the regional contribution, I would suggest making a clearer separation of the TOT AMA variable from the other percentages, as it would be otherwise confusing!*

Reply: The TOT variable is not actually a percentage. The conversion via the conversion factor needs to be used (for Fig. 10: 1% corresponds to 4000 trajectories). In the ACPD version we provided this separation via the light grey vertical dashed line. We made this line darker and doubled it and we made the separation clearer by adding a different axis to the right side of the plot.

*Line 262: Does it imply that the uplift is more intense in the TP and IND region, while the WP is contributing as much only because of the larger spatial extent of the defined region?*

Yes, concerning the uplift to the AMA we would say so.

*Page 21: this whole section can be summarized in a few sentences!*

As the Figs. 15 and 16 of the ACPD version have been combined in one panel, we had to revise the corresponding text of Fig. 16 (Fig. 14 b of the revised version) and made the description more concise.

*Discussion and Summary and conclusion:*
*Those two sections are also excessively verbose and with several repetitions be-*
*tween the two. I would suggest cleaning the text and really focus on the important*
*messages (for example the section 5.2 and 5.3 could be significantly shortened)*
*and avoid stating the same conclusion between sections 5 and 6.*
We shortened and/or cleaned up the respective sections. Further, as requested
by reviewer #1 and #2 we made the entire manuscript less repetitive. Hence,
some parts have been (re)moved from/to the discussion/summary.

*Technical comments:*

*Line 3: "analyses".*
Reply: Spelling corrected. Thank you!

*Line 3: in the same line there is the use of English and American notation.*
*Please correct!*
Reply: We are sorry, but we do not see where AE and BE are mixed. However,
we exchanged "we analyze" with "we investigate".

*Line 29: In the Asian summer monsoon (ASM) regions, the heating. . . .*
Reply: The wording has been changed to: "In the Asian summer monsoon
(ASM) region, the heating ...".

*Caption Figure 1: Better specify here how the TP contours are chosen rather*
*than on Figure 2.*
Reply: An explanation regarding the TP contour is now added. Further, the
contours have been modified (see your comment with respect to Line 255).

*Line 230: put a comma between "indicated above" and "the trajectories start*
*to fill"*
Reply: Done.

*Line 390: the comma after the "help to discern" can be removed.*
Reply: Done.

[revised manuscript text omitted]

---

## Editor Decision (ED1)

Editor decision for paper

acp-2022-143

**Climatology and variability of air mass transport from the boundary layer to the Asian monsoon anticyclone**

by M. Nützel et al.

I thank the authors for submitting their revised version. While reviewer 2 is happy with the revisions and accepts the paper for publication, reviewer 1 – a very experienced colleague – contacted me offline and indicated that "the authors largely rejected my comments and did not do much, if anything, that I suggested. I can't recall having a review treated this way before. I see two options: 1) to ask them to take my comments seriously and revise the paper, or 2) to publish as is. I am OK with option 2, although I think it leaves the paper much weaker than it needs to be. I will leave the decision up to you." While I understand that redoing the entire study with ERA5 would be an enormous task and beyond what can be done during revisions, the remark that the reviewer felt his/her comments to be largely ignored is problematic. I therefore had a closer look at the revised version, having in mind the general comment 6 from reviewer 1 about the conciseness of the writing. Although you shortened certain parts, the text is not yet fully reader friendly. I found several parts of the text unclear or distracting. Below my comments and suggestions.

L4: to me this sentence only makes sense if I insert hyphens "… displacements of the AMA with the PBL-to-AMA-transport". Is this what you intend to say? This term appears many times in the paper. If you prefer a formulation without hyphens then I would suggest "… with the transport from the PBL to the AMA".

L11: why "above"?

L15: why not simply "variability of PBL source regions"?

L34: you might like to add here a reference to the recent paper by Clemens et al. 2022:
Clemens, J., F. Ploeger, P. Konopka, R. Portmann, M. Sprenger, and H. Wernli, 2022. Characterization of transport from the Asian summer monsoon anticyclone into the UTLS via shedding of low potential vorticity cutoffs. Atmos. Chem. Phys., 22, 3841–3860.

L46: you often use "with respect to" when – in my view – a simpler construction would be much clearer, see also remark above. Here my suggestion would be "highlighted the importance of the Tibetan Plateau for the transport …" Please ask the native speakers in the team of authors to check the use of "with respect to" throughout the paper.

L55: "analysis" should read "analyses"

L78: not sure whether I understand this question. Do you mean "Are the PBL source regions and the transport pathways affected by / sensitive to interannual east-west shifts of the AMA?"

L82: no need for "In particular"

L83: this sentence does not work, maybe "Results from the Lagrangian model will serve for a comparison with …"

L97: I wonder whether the results of the study are sensitive to the choice of the starting level – here 150 hPa. This choice is not well motivated. Would you have trajectories at hand to check, whether a starting level of 100 or 200 hPa would lead to different results? At least you should better explain why this starting level is appropriate (and sufficient) to capture the entire transport from the PBL to the AMA.

L149: Is it correct that you use this GPHA threshold criterion only at 150 hPa? If yes, please mention this explicitly.

Section 2.3.1: I find it a bit painful to read this section. Please shorten the text, if I understand correctly, what you explain here is that you do not consider all years from 1979 to 2013, but only 14 years, and you selected them such as to capture the variability in the W-E position of the AMA as expressed by the South Asian High Index. This can be said in a few lines. And please list the 7 years each that were chosen for the west/east position of the AMA.

I don't think that Fig. 2 is needed in this paper. Vertical motion at 150 hPa is not very relevant for the transport from the PBL to this level.

L198: Just write "First, we investigate the climatological …"

L204: I think this is a very important point: you write here that you only consider trajectories that reach the PBL top within 90 days. How many of the AMA backward trajectories started at 150 hPa fulfill this criterion? I think it is important to mention this percentage. If it is substantially lower than 80%, then it might make sense to show Fig. 3 only for the trajectories that also fulfill the PBL criterion. Currently it is a bit strange that so many trajectories are started at 150 hPa over the Arabian Peninsula (Fig. 3), but this region does not appear at all when looking at the 200-hPa crossings (Fig. 4a). This should be discussed, and maybe the reason is that the Arabian Peninsula trajectories don't reach the PBL within 90 days(?).

L207: I don't understand "or below"

L209: I apologize but I am lost here. What you show in Fig. 4 are trajectories that fulfill at 150 hPa your AMA criterion and that cross the PBL height within 90 days (backward in time). I don't understand what you mean by "not necessarily the full three-dimensional pathways", do you mean by this, e.g., trajectories that don't go back to the PBL or trajectories that rise up only to 200 but not 150 hPa? And I am totally at lost with understanding why you show Fig. 5. Why is Fig. 5a so totally different from Fig. 4a? Also, the distinction between trj1 and trj2 does not seem very relevant to me. Things should be clear if you write that Fig. 4 shows the last upward crossing of the XX hPa level. (I assume that if a trajectory crosses a certain pressure level more than once, you only retain one crossing?).

L218: Here you write "To get a better picture of the full transport pathways …", which is now confusing after Fig. 5. Do you now continue with the trajectories shown in Fig. 4, or does "full transport" mean that you include here other trajectories as well?

I wonder whether the results in Figs. 4 and 6 are fully consistent. Fig. 4a shows no 200-hPa crossings west of 60°E, whereas Fig. 6d shows many trajectories west of 60°E at altitudes from 9-15 km. Please discuss this discrepancy or my misunderstanding when comparing the two figures. And as noted by one of the reviewers, it would be most helpful to have a pressure axis in Fig. 6 (e.g., to make a good comparison with Fig. 4). This would be much more reader-friendly than the barometric height formula and the complicated text in L219.

L238: where can the reader see the "upward circling"? I don't doubt that this interpretation is correct, but I don't see it in the results shown. My understanding of upward circling is that the trajectories follow a circular path in the horizontal from the PBL to the AMA at 150 hPa. But the panels in Fig. 4 allow all sorts of interpretations of how air parcels, e.g., from the Northern Philippines move from the PBL to 150 hPa. This ascent could also be rather vertical according to Fig. 4, so what does "circling" mean?

L239: typo in "refines"

L256: very complicated "on individual dates with respect to the initialization date". I assume that Fig. 9 is done in the same way as Fig. 6?

L259: "remain stable" is not clear enough, what you mean is that the PBL sources seem to be very similar in years with an eastern position of the AMA vs. years with a western position of the AMA.

L267: Oh, now you quantify for the first time the trajectories that do not cross the PBL within 90 days (see my comment above)! I don't find it ideal that now, in Fig. 10, you consider all trajectories started from the AMA, whereas Figs. 4-9 only considered those that crossed the PBL. Therefore, the percentages in Fig. 10 do not correspond to percentages of trajectories shown, e.g., in Fig. 4. It

would be more reader-friendly, if the noX trajectories were mentioned earlier in the paper (before Fig. 4) and from then on, only PBL-crossing trajectories were considered.

L273-284: I suggest omitting this analysis, because it is already clear from Figs. 8b and 9 that the east-west position of the AMA does not matter for the PBL source regions.

L301: "transport from the TP into the AMA occurs vertically" – how does this correspond to the "upward circling" mentioned before?

L304: no need to motivate here again the need to look at intraseasonal variability, as you already discussed this in the previous subsection!

L309 and Figure 14: again, it is not ideal / not necessary that the noX trajectories are included.

L339: To me a supplement is a separate document, but you include Fig. B2 in an appendix, which is part of the main paper. Please decide about your strategy and terminology.

L363-366: I suggest omitting this short paragraph, because the reader does not really understand how you varied the PBL identification, and it is a bit arbitrary to test this sensitivity for LG-D but not for TRJ.

Discussion of Fig. 17: I am a bit confused why now hemispheric results are shown and the discussion includes the North American monsoon. Why not confine the analysis and discussion to the main theme of the paper?

L427-435: Why introducing here a discussion about the MHI? I think it is one conclusion that the west-east position of the AMA does not influence the upward transport substantially, so why then adding an excursion about MHI and SAHI?

L454: I would not dare to make such a statement. All data sets used so far for studying the transport into the AMA are far away from convection-resolving simulations. I think we need such simulations to really assess the impact of deep convection.

L500: "However, we found an upward circling already considerably below 150 hPa for approximately half of the PBL crossing trajectories." I don't understand where this result has been shown in this study.

L501 "The attribution of PBL source regions, however, is less clear" – what do you mean by less clear? Do you mean are more sensitive to the model / approach used? Or do you mean that a large set of source regions contributes?

Finally, I would like to briefly comment on your replies to the general comments of reviewer 1. While I can follow your argumentation about the difference between PBL source distributions and precipitation – indeed, you "only" look at upward transport that reaches 150 hPa – I thought that you might be able to do some sensitivity tests with your TRJ approach using hours ERA5 data. You mention that doing this for the entire study would be a huge effort. I fully agree. But already backward trajectories from 150 hPa for a single JJA season with hourly ERA5, 6-hourly ERA5 and 6-hourly ERA-Interim would be tremendously insightful. I cannot estimate how difficult it is for you to do such an analysis, and therefore I leave it up to you whether you include it or not in the final version of your paper. And about the complex link of the source maps and precipitation: could it make sense to discuss this in your discussion section? I find it interesting that with a starting level of 150 hPa, one obviously "misses" a lot of the vertical transport in the monsoon region associated with intense precipitation.

---

## Author Response (AR2)

**Author Comment to the revised version of manuscript ACP-2022-143 (https://doi.org/10.5194/acp-2022-143, in review, 2022): "Climatology and variability of air mass transport from the boundary layer to the Asian monsoon anticyclone" (revised title)**

by M. Nützel et al.

October 17, 2022

We thank the referees for taking time to reevaluate our revised paper. In particular, we also thank the editor for taking the time to read our manuscript and provide detailed and helpful suggestions/comments to improve the paper. Both, the previous reviews and the current editor comments are very much appreciated. In the following we address each comment of the editor (*black italics*) by stating our reply (blue). In addition we append a manuscript version which highlights the changes between the revised version and the current manuscript version, i.e. the version after the second revision.

**Reply to editor comments**

Below we will address all comments of the editor and we will state corresponding changes in the manuscript. Again, we would like to thank the editor for taking the time to comment on our revised manuscript.

*Editor decision for paper*

*acp-2022-143*

**Climatology and variability of air mass transport from the boundary layer to the Asian monsoon anticyclone**

*by M. Nützel et al.*

*I thank the authors for submitting their revised version. While reviewer 2 is happy with the revisions and accepts the paper for publication, reviewer 1 – a very experienced colleague – contacted me offline and indicated that "the authors largely rejected my comments and did not do much, if anything, that I suggested. I can't recall having a review treated this way before. I see two options: 1) to ask them to take my comments seriously and revise the paper, or 2) to publish as is. I am OK with option 2, although I think it leaves the paper much weaker than it needs to be. I will leave the decision up to you." While I understand that redoing the entire study with ERA5 would be an enormous task and beyond what can be done during revisions, the remark that the reviewer felt his/her comments to be largely ignored is problematic. I therefore had a closer look at the revised version, having in mind the general comment 6 from reviewer 1 about the conciseness of the writing. Although you shortened certain parts, the text is not yet fully reader friendly. I found several parts of the text unclear or distracting. Below my comments and suggestions.*

We greatly appreciate the editor's thoughtful comments and suggestions, which we will address below. We have revisited the point from Reviewer 1 with respect to the sensitivity of using ERA-Interim vs ERA5 to calculate the trajectories. Although we have considered to re-do the calculation for one season using ERA5, the decision in the end was to discuss the expected changes if using ERA5, based on some new diagnostics performed by co-author Laura Pan's group. These diagnostics go beyond showing differences in trajectory model studies driven by

ERA-Interim vs ERA5. The vertical wind products and the trajectory results are evaluated using two observation-based diagnostics. This discussion is now included in revised Section 5.2, cited here:

"The representation of convective transport in the trajectory analyses forms the leading uncertainty in our results. This uncertainty can be addressed with two related questions: 1) how well is convective transport represented in trajectory analysis, which use the resolved winds of analysis products? 2) What is the sensitivity of the calculations to the analysis products used? In particular, what is the influence of the relatively coarse spatial and temporal resolution of the ERA-Interim data employed in this study (here 1.5° and 6 hourly) on the presented results versus that of the newer generation reanalysis ERA5 (Hersbach et al., 2020) at high horizontal resolution ($\sim$0.25°), provided in hourly intervals?

These questions are examined in a recent work by Smith et al. (2021), in which convective transport time scales were quantitatively characterized using transit time distributions (TTDs), analogous to the age spectra, or distributions of the age of air, in stratospheric transport studies (e.g. Hall and Plumb, 1994). The work uses a set of diagnostics to quantify the representation of convective transport in trajectory calculations, specifically, by comparing TTDs from trajectory model results with the chemical lifetime-based TTDs derived from airborne in situ measurements over the convection dominated Western Pacific. Four sets of wind products from commonly used operational analyses and re-analyses are examined in this study, including ERA-Interim and ERA5. The results of the study indicate that the trajectory-based TTD from ERA5 has comparable mode and mean to that of the chemical-lifetime based TTD. The ERA-Interim based TTD on the other hand, shows considerably slower transport, although showing qualitatively similar distribution in transport origins at the boundary layer. Using the TTD diagnostic, the ERA-Interim based calculation misses approximately 30% of the convective transport (Smith et al., 2021, Table 2).

Based on this diagnosis, we expect that if the higher spatial and

temporal resolution products from ERA5 were used, the result of this study would show enhanced convective transport which should lead to a higher percentage of back-trajectories that reach the top of the PBL within the season. This assessment is also in agreement with the presented EMAC-ATTILA data, which contain the effect of parametrized convection and show an higher fraction of young (<90 days) air masses in the AMA than the TRJ data (Fig. 14). Further the EMAC-ATTILA data also support key characteristics of the transport pathways and the increasing contribution of the TP to AMA air masses over the course of the monsoon season. For the distribution of PBL source regions, although we expect changes in detail, the overall conclusions in the large-scale perspective are not expected to change. The latter is also supported by Legras and Bucci (2020), who show similar source regions based on ERA5 (and ERA-Interim data) with an entirely different modelling approach (i.e. a combination of reanalysis and observational data)."

The statement regarding the PBL source regions is also supported by a more recent paper, Pan et al., in revision (JGR, minor revision), where the time scales and contributing boundary layer of Asian Monsoon transport over the Western Pacific are calculated using a trajectory model, which is driven by ERA5 for one season. This - yet to be published - new result serves as an update to the published 39-year climatology based on ERA-Interim (Honomichl and Pan, 2020). In this case, the ERA5 result is consistent with the ERA-Interim in the large-scale perspective, although the ERA5 result provides much better details in the distribution of contributing boundary layer. Based on these studies (Smith et al., 2020, and Pan et al., in revision), we expect that a re-do of the study using ERA5 would add significant more insight into the transport process, but it would be "an enormous task and beyond what can be done during revisions", as remarked by the editor. Further, even a single season intercomparison would need additional experiments e.g. as in (Hoffmann et al., 2019) to provide context and would clearly shift the focus of the paper. We hope the discussion in the revised Section 5.2 provides sufficient information for the readers to put the presented results into perspective.

*L4: to me this sentence only makes sense if I insert hyphens "... displace-ments of the AMA with the PBL-to-AMA-transport". Is this what you intend to say? This term appears many times in the paper. If you prefer a formulation without hyphens then I would suggest "... with the transport from the PBL to the AMA".*

Yes, that is what we wanted to convey. We have checked the entire manuscript and changed the respective phrase to either the first (including hyphens) or second suggestion (no hyphens) of the editor.

*L11: why "above"?*

The sentence was adapted. Please see our reply to your comment L207 concerning "below".

*L15: why not simply "variability of PBL source regions"?*

Changed.

*L34: you might like to add here a reference to the recent paper by Clemens et al. 2022: Clemens, J., F. Ploeger, P. Konopka, R. Portmann, M. Sprenger, and H. Wernli, 2022. Characterization of transport from the Asian summer monsoon anticyclone into the UTLS via shedding of low potential vorticity cut-offs. Atmos. Chem. Phys., 22, 3841–3860.*

The reference was added. Thank you for pointing it out.

*L46: you often use "with respect to" when – in my view – a simpler con-struction would be much clearer, see also remark above. Here my suggestion would be "highlighted the importance of the Tibetan Plateau for the transport ..." Please ask the native speakers in the team of authors to check the use of "with respect to" throughout the paper.*

Corrected here. We have checked the entire manuscript for "with respect to" and also for "via" and largely replaced the phrases.

*L55: "analysis" should read "analyses"*

Corrected. Thank you.

*L78: not sure whether I understand this question. Do you mean "Are the PBL source regions and the transport pathways affected by / sensitive to inter-annual east-west shifts of the AMA?"*

Yes, this is what is meant. The very simple question would be: If the AMA is located rather to the east or west, do we see any differences in the pathways or/and source regions? We chose the wording "related to" on purpose as the east–west shifts of the AMA might not be what is causing the different contributions in the first place but might rather be themselves a response to changed heating, i.e. we did not want to imply a causal relationship. We changed the wording to "sensitive to" - hoping that this does not suggest a causal relation, while keeping "related to" at a few other instances in the text.

*L82: no need for "In particular"*

Removed. The sentence now starts with: "These Lagrangian CCM results ..."

*L83: this sentence does not work, maybe "Results from the Lagrangian model will serve for a comparison with ..."*

We rephrased the sentence to: "Results from the Lagrangian model will help to assess the sensitivity of the results to the modelling approach as..."

*L97: I wonder whether the results of the study are sensitive to the choice of the starting level – here 150 hPa. This choice is not well motivated. Would you have trajectories at hand to check, whether a starting level of 100 or 200 hPa would lead to different results? At least you should better explain why this starting level is appropriate (and sufficient) to capture the entire transport from the PBL to the AMA.*

We added a paragraph to explain the choice of the starting level of the trajectories:

> "We chose the 150 hPa level to initialize the trajectories as it roughly corresponds to the 360 K from which trajectories tend to further ascend into the stratosphere (Garny and Randel, 2016). Moreover, the 150 hPa level is a level where we find strong anticyclonic circulation based on the maximum and minimum zonal wind speeds in the UT in the Asian monsoon region (see e.g. Fig. 1 of Garny and Randel, 2016). From the analysis shown in (Bergman et al., 2013) for the 100 and 200 hPa level, we expect that our qualitative results are not strongly dependent on the choice of the starting level."

*L149: Is it correct that you use this GPHA threshold criterion only at 150 hPa? If yes, please mention this explicitly.*

For TRJ the threshold is used only at 150 hPa to select the AMA trajectories from all trajectories initialized at 150 hPa. But for LG data as a starting range is used (140-160 hPa) the criterion is applied for this pressure range together with a restriction on the longitude/latitude to filter out trajectories that start within the AMA. We have updated the respective text and explicitly added the sentence: "We emphasize that the GPHA criterion is only applied once at the starting point of the trajectories or air parcels to determine whether they are located within the AMA."

*Section 2.3.1: I find it a bit painful to read this section. Please shorten the text, if I understand correctly, what you explain here is that you do not consider all years from 1979 to 2013, but only 14 years, and you selected them such as to capture the variability in the W-E position of the AMA as expressed by the South Asian High Index. This can be said in a few lines. And please list the 7 years each that were chosen for the west/east position of the AMA.*

Yes, your understanding is correct. We have shortened the respective section. For further information on the SAHI (as requested by referee 2) and the list of the selected summer seasons we refer the reader to the Appendix: "The selected summer seasons are listed in the Appendix A2, where also a description of the modified SAHI and of the selection process is presented."

*I don't think that Fig. 2 is needed in this paper. Vertical motion at 150 hPa is not very relevant for the transport from the PBL to this level.*

We have checked the differences also at 175 hPa (and also at 200 hPa) and the differences look (relatively) similar to the differences at 150 hPa. Previously, we showed this figure to motivate the choice of the 14 summer seasons. Nevertheless, as suggested we have removed this figure and the motivation for the selection is now entirely by referring to Fig. 14 of Nützel et al. (2016).

*L198: Just write "First, we investigate the climatological ..."*

Changed as suggested. To increase the readability of the paper, we checked the entire manuscript and tried to shorten/adapt introductory clauses, where appropriate.

*L204: I think this is a very important point: you write here that you only consider trajectories that reach the PBL top within 90 days. How many of the AMA backward trajectories started at 150 hPa fulfill this criterion? I think it*

232 *is important to mention this percentage. If it is substantially lower than 80%,*
233 *then it might make sense to show Fig. 3 only for the trajectories that also fulfill*
234 *the PBL criterion. Currently it is a bit strange that so many trajectories are*
235 *started at 150 hPa over the Arabian Peninsula (Fig. 3), but this region does not*
236 *appear at all when looking at the 200-hPa crossings (Fig. 4a). This should be*
237 *discussed, and maybe the reason is that the Arabian Peninsula trajectories don't*
238 *reach the PBL within 90 days(?).*

239 We agree that it is helpful to early state the fraction of AMA trajectories that
240 reach the PBL. Hence we added a sentence in Sect. 3.1: "For the analysis of
241 the transport pathways, we will only consider trajectories that start within the
242 AMA and reach the PBL within 90 days, whereas in the analyses of the PBL
243 sources we also quantify the fraction of trajectories starting within the AMA
244 that do not reach the PBL within 90 days (roughly 15%, see Sect. 3.1.2)." Al-
245 though it can be assumend from the large fraction that reaches the PBL within
246 90 days ($\sim$85%), we have explicitly checked that Fig. 3 does not change sub-
247 stantially if only PBL crossing trajectories are considered.

248 The trajectory starts at 150 hPa are simply related to the occurence of the AMA
249 in the respective region. As the AMA spans also to the Arabian peninsula (see
250 e.g. Fig. 4 in Nützel et al., 2016), the start of trajectories at 150 hPa in this region
251 are correct. Further, a trajectory started at 150 hPa over the Arabian Penin-
252 sula, does not have to vertically ascend to that level from the Arabian peninsula.
253 The trajectories are indeed transported upward over the south-eastern side of
254 the AMA as can be seen in our Fig. 4, which is in agreement with the findings
255 of (Bergman et al., 2013). We think that additional clarification of that issue
256 will be given in our replies to your comments L209 and L218.

258     *L207: I don't understand "or below"*

259 As the AMA starts only at a certain height level we wanted to be clear that the
260 upward transport is on the south-eastern side of the AMA where it exist and
261 below the south-eastern part of the AMA in the height region, where the AMA
262 does not exist. However, we can understand that this distinction is confusing, so
263 we rephrased to try to make our statement clearer: "With increasing height, the
264 upward transport of air masses focuses on (the region below) the south-eastern
265 part of the AMA."

267     *L209: I apologize but I am lost here. What you show in Fig. 4 are trajecto-*
268 *ries that fulfill at 150 hPa your AMA criterion and that cross the PBL height*

*within 90 days (backward in time). I don't understand what you mean by "not*
*necessarily the full three-dimensional pathways", do you mean by this, e.g., tra-*
*jectories that don't go back to the PBL or trajectories that rise up only to 200*
*but not 150 hPa? And I am totally at lost with understanding why you show*
*Fig. 5. Why is Fig. 5a so totally different from Fig. 4a? Also, the distinction*
*between trj1 and trj2 does not seem very relevant to me. Things should be clear*
*if you write that Fig. 4 shows the last upward crossing of the XX hPa level. (I*
*assume that if a trajectory crosses a certain pressure level more than once, you*
*only retain one crossing?).*

There is still a misunderstanding and it seems that we have not been clear enough with our explanation. With the "full 3-d pathways" we do not mean a different subset of the trajectories, i.e. we still analyze trajectories that start in the AMA at 150 hPa and reach the PBL within 90 days. And yes, only the final crossing points are retained in Figs. 4, 8 and 11. To avoid any misunderstanding we replaced the phrase "full pathways" (or similar) and we noted in the discussion of Fig. 4 that only the final crossing point is registered.

Figs. 5a and 5b were actually meant to clarify why the analyses e.g. in Figs. 4 can not be used to infer that trajectories are only located in these regions on their way to the 150 hPa level in the AMA. Fig. 4 and Fig. 5 a are different as they depict the upward and downward crossings of trajectories at ∼200 hPa. As on a climatological basis on the east side of the AMA upward winds are present and on the west side downward winds (e.g. Nützel et al., 2016, their Fig. 10), upward crossings as they are diagnosed in Fig. 4 are most likely to be detected in this region. Hence, although trajectories might be located also at different horizontal positions (e.g. in the western part of the AMA, as seen in the starts of the TRJ or the density distributions Fig. 6 etc.) they will only be noted on the eastern side in analysis of Fig. 4 as this is the region where they are transported upwards. The "snapshots" of the location of the trajectories 1, 2.5 , 5 and 15 days prior to their starting date (Fig. 6) do not exhibit this "flaw" and hence trajectories that circle within the AMA and are located on the western side at the time of the snapshot are noted as well. In contrast, Fig. 5a reverses the analysis in Fig. 4 and looks where trajectories experience downward movement and hence here (in agreement with the location of downward movement on the western side of the AMA) the western side of the AMA shows up in this analysis. The hypothetical trajectories trj1 and trj2 are different as trj1 experiences upward and downward motion close to the 200 hPa level, whereas trj2 simply continues to further rise after crossing the 200 hPa level: the final crossing points

of the 200 hPa level of both trajectories are registered in Fig. 4 - and actually all PBL-crossing trajectories are registered once in Fig. 4 as they somehow have to cross the 200 hPa level on their way from the PBL to the 150 hPa level. As only trj1 experiences the upward/downward transport around the 200 hPa level, only trj1 is noted in Fig. 5a which displays the regions of downward transport of trajectories. To facilitate the understanding, we adjusted the figure captions of Figs. 4 and 5a, so the terminology "upward" and "downward" crossing are easier to spot. Further, we adjusted the discussion of Fig. 5a (now Fig. 4a).

*L218: Here you write "To get a better picture of the full transport pathways ...", which is now confusing after Fig. 5. Do you now continue with the trajectories shown in Fig. 4, or does "full transport" mean that you include here other trajectories as well? I wonder whether the results in Figs. 4 and 6 are fully consistent. Fig. 4a shows no 200-hPa crossings west of 60°E, whereas Fig. 6d shows many trajectories west of 60°E at altitudes from 9- 15 km. Please discuss this discrepancy or my misunderstanding when comparing the two figures. And as noted by one of the reviewers, it would be most helpful to have a pressure axis in Fig. 6 (e.g., to make a good comparison with Fig. 4). This would be much more reader-friendly than the barometric height formula and the complicated text in L219.*

The subset of trajectories does not change between Figs. 4, 5a and 6. The wording has been adjusted to avoid any confusion. We hope that our reply to your comment L209, clarifies that there is no inconsistency, as the difference between Figs. 4 and 6 is entirely caused by the underlying analysis method. Fig. 4 depicts the locations of the final upward crossings of trajectories through a specific surface - hence only regions and time steps where trajectories experience upward motion are noted. In contrast Fig. 6 does not make such a restriction, but simply shows a snapshot of the trajectories on their pathway to the 150 hPa level. We included the pressure axis in the respective plots to facilitate the intercomparison. We did not do that before as the figures (in the multi-panel) get smaller and as the units of the displayed quantities are given with respect to the log-pressure height. Further, we adapted the figure captions and tried to shorten the explanation in L219.

*L238: where can the reader see the "upward circling"? I don't doubt that this interpretation is correct, but I don't see it in the results shown. My understanding of upward circling is that the trajectories follow a circular path in*

the horizontal from the PBL to the AMA at 150 hPa. But the panels in Fig. 4 allow all sorts of interpretations of how air parcels, e.g., from the Northern Philippines move from the PBL to 150 hPa. This ascent could also be rather vertical according to Fig. 4, so what does "circling" mean?

Maybe this is a misunderstanding. The upward circling was meant to occur only after the trajectories have been transported to a certain height/pressure level: so there is first vertical transport and then recirculation within the AMA (with downward/upward transport on the western/eastern side of the AMA). This is supported by our analyses (Figs. 4, 5a and 6) and is in agreement with the findings of Vogel et al. (2019) and Legras and Bucci (2020) (and to some extent with the study by Bergman et al. (2013)). Whether the net circulation is upward or not, we cannot deduce with our analysis and hence when it comes to our study, we rephrased the term "upward circling".

L239: typo in "refines"

Corrected.

L256: very complicated "on individual dates with respect to the initialization date". I assume that Fig. 9 is done in the same way as Fig. 6?

Yes, Fig. 9 shows the corresponding differences of Fig. 6 for west minus east years. We agree that it was difficult to follow, hence we revised the respective sentence: "To capture the differences of the trajectory pathways between years with a rather western and rather eastern position of the AMA, Fig. 9 shows the corresponding composite differences (west minus east) of the analyses in Fig. 6."

L259: "remain stable" is not clear enough, what you mean is that the PBL sources seem to be very similar in years with an eastern position of the AMA vs. years with a western position of the AMA.

Yes, concerning the pathways that is what we meant. Hence we rephrased to: "Overall, there are no qualitative differences in the transport pathways between years with a rather eastward and years with a rather westward location of the AMA."

L267: Oh, now you quantify for the first time the trajectories that do not cross the PBL within 90 days (see my comment above)! I don't find it ideal that now, in Fig. 10, you consider all trajectories started from the AMA, whereas Figs. 4-9 only considered those that crossed the PBL. Therefore, the percentages

*in Fig. 10 do not correspond to percentages of trajectories shown, e.g., in Fig. 4.*
*It would be more reader-friendly, if the noX trajectories were mentioned earlier*
*in the paper (before Fig. 4) and from then on, only PBL-crossing trajectories*
*were considered.*

In response to your comment on L204, we have added a clarification that for the
transport pathways only PBL-crossing trajectories are analysed in Sect. 3.1. We
hope that this clarification helps the reader to follow the manuscript more eas-
ily. We agree that the percentages (if one would integrate e.g. Fig. 4 at 0.85*ps)
would not match with the percentage given in Fig. 10. However, we think that
the information on the noX trajectories is valuable - as we would also guess
from your comment L204 - and decided that it should not be removed from the
plots. Hence, we also though about reversing the appearance/discussion of the
plots. That would mean to first discuss Fig. 10 and then Figs. 4-9 etc. However,
accordingly also Figs. 11-13 and 14 would need to be switched, leading to the
problem that the reader would have to switch between different "bases" (AMA
vs AMA and PBL-crossing) again. Thus, instead of removing the noX trajecto-
ries or reversing the appearance, we worked on that issue by being more precise
which subset is being analysed, e.g. by updating Sect. 3.1. and by explicitly
mentioning the noX trajectories in the discussion of the respective analyses.

*L273-284: I suggest omitting this analysis, because it is already clear from*
*Figs. 8b and 9 that the east-west position of the AMA does not matter for the*
*PBL source regions.*

We agree that this is the case for the mean, however the interannual variation
and also additional information, e.g. concerning the total number of trajectories
and the fraction of the noX trajectories can not be inferred from Figs. 8 and
9. Hence we made the compromise to keep the figure while we substantially
shortened this paragraph.

*L301: "transport from the TP into the AMA occurs vertically" – how does*
*this correspond to the "upward circling" mentioned before?*

First we hope that it is now clearer, that the recirculation is meant to take
place only after a first "vertical uplift" (see our reply to your comment L238).
Still, we thank the editor for spotting this unclear statement. What is exactly
meant is, that in June air masses, which are transported vertically above the
TP eventually encounter the STJ (typically at levels below 150 hPa) and get
advected out of the monsoon region - hence they cannot contribute to the AMA

air masses at 150 hPa. In August this is different as air masses transported from the TP can either ascend vertically up to 150 hPa or get entrained into the AMA circulation at some level and from there circle to the 150 hPa level (of course not all air masses from the TP but these are the ones that get noted in our analyses). Hence we rephrased to: "In August the AMA is located above the TP and air masses from the TP can directly feed into the core of the AMA."

*L304: no need to motivate here again the need to look at intraseasonal variability, as you already discussed this in the previous subsection!*
The motivation was deleted here and the motivation in Sect. 3.1. was revised.

*L309 and Figure 14: again, it is not ideal / not necessary that the noX trajectories are included.*
Please consider our comments to your comments concerning L204 and L267.

*L339: To me a supplement is a separate document, but you include Fig. B2 in an appendix, which is part of the main paper. Please decide about your strategy and terminology.*
This is also our understanding. We would have split the documents in the end. But as the supplement is not long we thought it is more convenient for the review process to have all data in one document. Consequently, we decided to keep the figures in an Appendix called "Supporting figures".

*L363-366: I suggest omitting this short paragraph, because the reader does not really understand how you varied the PBL identification, and it is a bit arbitrary to test this sensitivity for LG-D but not for TRJ.*
As suggested we deleted the paragraph.

*Discussion of Fig. 17: I am a bit confused why now hemispheric results are shown and the discussion includes the North American monsoon. Why not confine the analysis and discussion to the main theme of the paper?*
We want to show where TP trajectories are located in the UT in June vs. August at various pressure levels to show the possibly stronger confinement/dispersion of TP trajectories. Hence, we need to make the analysis globally and such an analysis is only possible using the LG data and can not be done with the existing TRJ data. As can be inferred from Fig. 17 in August compared to June the TP trajectories are more likely to be located in the Asian monsoon region (stronger

confinement), whereas in June compared to August, the trajectories are dispersed more strongly and located downstream of the Asian monsoon region. This result corroborates the results from the TRJ data about a key process, which causes the different contributions of the TP in June vs. August. As the difference of the probability densities shows a local minimum in the North American monsoon region, we simply stated this in the text to explain this feature.

*L427-435: Why introducing here a discussion about the MHI? I think it is one conclusion that the west-east position of the AMA does not influence the upward transport substantially, so why then adding an excursion about MHI and SAHI?*

You are right, the east-west position has no substantial impact on the pathways or PBL sources. The reason to include here the discussion on SAHI and MHI is to motivate that it is likely that there are no dependencies/differences of the PBL sources etc. if stratifying/compositing against/with the MHI. We think, that so far the connection between MHI and SAHI has not been analyzed.

*L454: I would not dare to make such a statement. All data sets used so far for studying the transport into the AMA are far away from convection-resolving simulations. I think we need such simulations to really assess the impact of deep convection.*

We fully agree and we thank the editor for spotting this unclear statement. Hence, we went through the manuscript and rephrased to only address parametrized convection with such statements. The sentence at hand was actually removed during the revision of Sect. 5.2.

*L500: "However, we found an upward circling already considerably below 150 hPa for approximately half of the PBL crossing trajectories." I don't understand where this result has been shown in this study.*

Concerning the use of the phrase "upward circling", we refer to our reply on your comment L238. Further, in the discussion of Fig. 5a (now Fig. 4a), which shows the regions of downward crossing, we note that approximately 50% of the PBL-crossing trajectories are noted in the respective analysis. From this we infer that roughly 50% of the trajectories recirculate within the AMA considerably below the starting level.

*L501 "The attribution of PBL source regions, however, is less clear" – what do you mean by less clear? Do you mean are more sensitive to the model / approach used? Or do you mean that a large set of source regions contributes?* We rephrased the sentence and also added a colon, to indicate that the explanation follows. "The attribution of the PBL source regions, however, is less clear as it is more sensitive to the modelling approach: In TRJ, ..."

*Finally, I would like to briefly comment on your replies to the general comments of reviewer 1. While I can follow your argumentation about the difference between PBL source distributions and precipitation – indeed, you "only" look at upward transport that reaches 150 hPa – I thought that you might be able to do some sensitivity tests with your TRJ approach using hours ERA5 data. You mention that doing this for the entire study would be a huge effort. I fully agree. But already backward trajectories from 150 hPa for a single JJA season with hourly ERA5, 6-hourly ERA5 and 6-hourly ERA-Interim would be tremendously insightful. I cannot estimate how difficult it is for you to do such an analysis, and therefore I leave it up to you whether you include it or not in the final version of your paper. And about the complex link of the source maps and precipitation: could it make sense to discuss this in your discussion section? I find it interesting that with a starting level of 150 hPa, one obviously "misses" a lot of the vertical transport in the monsoon region associated with intense precipitation.*

We agree that including a discussion concerning the differences between precipitation maps and source maps makes sense. We have added a paragraph about this issue in the discussion Sect. 5.1, which is based on our previous reply to reviewer 1. This paragraph explains the seeming inconsistency between precipitation maps and source regions of AMA air masses. Actually, we debated about including such a discussion in the last version (first revision, 13 July) of our paper, however, previously we decided otherwise to shorten the manuscript.

The sensitivity of our results with respect to ERA-Interim has been addressed in the revised Section 5.2. Please, see also our reply to your first comment. According to the method of quantifying convective transport in Table 2 of Smith et al. (2021), ERA-Interim missed ∼30% of convective transport in the trajectory model experiment over the Western Pacific. A single season sensitivity test is done in the work of Pan et al., (JGR in minor revision) for a very similar problem (see Honomichl and Pan, 2020, where the boundary layer and transit time of air mass transported from the AMA to Western Pacific are quantified using ERA-I driven kinematic back trajectory). The result shows similar spatial pattern of PBL encounter in the large scale but with more details along the monsoon trough.

**References**

J. W. Bergman, F. Fierli, E. J. Jensen, S. Honomichl, and L. L. Pan. Boundary layer sources for the Asian anticyclone: Regional contributions to a vertical conduit. *J. Geophys. Res.-Atmos.*, 118(6):2560–2575, 2013. ISSN 2169-8996. doi: 10.1002/jgrd.50142. URL http://dx.doi.org/10.1002/jgrd.50142.

H. Garny and W. J. Randel. Transport pathways from the Asian monsoon anticyclone to the stratosphere. *Atmospheric Chemistry and Physics*, 16 (4):2703–2718, 2016. doi: 10.5194/acp-16-2703-2016. URL http://www.atmos-chem-phys.net/16/2703/2016/.

T. M. Hall and R. A. Plumb. Age as a diagnostic of stratospheric transport. *Journal of Geophysical Research: Atmospheres*, 99(D1):1059–1070, 1994. doi: https://doi.org/10.1029/93JD03192. URL https://agupubs.onlinelibrary.wiley.com/doi/abs/10.1029/93JD03192.

H. Hersbach, B. Bell, P. Berrisford, S. Hirahara, A. Horányi, J. Muñoz-Sabater, J. Nicolas, C. Peubey, R. Radu, D. Schepers, A. Simmons, C. Soci, S. Abdalla, X. Abellan, G. Balsamo, P. Bechtold, G. Biavati, J. Bidlot, M. Bonavita, G. De Chiara, P. Dahlgren, D. Dee, M. Diamantakis, R. Dragani, J. Flemming, R. Forbes, M. Fuentes, A. Geer, L. Haimberger, S. Healy, R. J. Hogan, E. Hólm, M. Janisková, S. Keeley, P. Laloyaux, P. Lopez, C. Lupu, G. Radnoti, P. de Rosnay, I. Rozum, F. Vamborg, S. Villaume, and J.-N. Thépaut. The ERA5 global reanalysis. *Quarterly Journal of the Royal Meteorological Society*, 146(730):1999–2049, 2020. doi: https://doi.org/10.1002/qj. 3803. URL https://rmets.onlinelibrary.wiley.com/doi/abs/10.1002/qj.3803.

L. Hoffmann, G. Günther, D. Li, O. Stein, X. Wu, S. Griessbach, Y. Heng, P. Konopka, R. Müller, B. Vogel, and J. S. Wright. From era-interim to era5: the considerable impact of ecmwf's next-generation reanalysis on lagrangian transport simulations. *Atmospheric Chemistry and Physics*, 19(5):3097–3124, 2019. doi: 10.5194/acp-19-3097-2019. URL https://acp.copernicus.org/articles/19/3097/2019/.

S. B. Honomichl and L. L. Pan. Transport from the asian summer monsoon anticyclone over the western pacific. *Journal of Geophysical Research: Atmospheres*, 125(13):e2019JD032094, 2020. doi: 10.1029/

2019JD032094. URL `https://agupubs.onlinelibrary.wiley.com/doi/` `abs/10.1029/2019JD032094`. e2019JD032094 2019JD032094.

B. Legras and S. Bucci. Confinement of air in the Asian monsoon anticyclone and pathways of convective air to the stratosphere during the summer season. *Atmospheric Chemistry and Physics*, 20(18):11045–11064, 2020. doi: 10.5194/acp-20-11045-2020. URL `https://acp.copernicus.` `org/articles/20/11045/2020/`.

M. Nützel, M. Dameris, and H. Garny. Movement, drivers and bimodality of the south asian high. *Atmos. Chem. Phys.*, 16(22):14755–14774, 2016. doi: 10.5194/acp-16-14755-2016. URL `http://www.atmos-chem-phys.net/16/` `14755/2016/`.

W. P. Smith, L. L. Pan, S. B. Honomichl, S. M. Chelpon, R. Ueyama, and L. Pfister. Diagnostics of convective transport over the tropical western pacific from trajectory analyses. *Journal of Geophysical Research: Atmospheres*, 126(17):e2020JD034341, 2021. doi: https://doi.org/10.1029/ 2020JD034341. URL `https://agupubs.onlinelibrary.wiley.com/doi/` `abs/10.1029/2020JD034341`. e2020JD034341 2020JD034341.

B. Vogel, R. Müller, G. Günther, R. Spang, S. Hanumanthu, D. Li, M. Riese, and G. P. Stiller. Lagrangian simulations of the transport of young air masses to the top of the Asian monsoon anticyclone and into the tropical pipe. *Atmospheric Chemistry and Physics*, 19(9):6007–6034, 2019. doi: 10.5194/acp-19-6007-2019. URL `https://acp.copernicus.org/articles/` `19/6007/2019/`.